# Improving Neural Architecture Search by Minimizing Worst-Case Validation Loss

## Abstract

Neural architecture search (NAS) aims at automatically searching for high-performance architectures and has achieved considerable progress. Existing NAS methods learn architectures by minimizing average-case validation losses. As a result, the searched architectures are less capable of making correct predictions under worst-case scenarios. To address this problem, we propose a framework which leverages a deep generative model to generate adversarial validation examples to measure the worst-case validation performance of an architecture and improves the architecture by minimizing the loss on the generated adversarial validation data. Our framework is based on multi-level optimization, which performs multiple learning stages end-to-end. Experiments on a variety of datasets demonstrate the effectiveness of our method.

## 1 Introduction

Neural architecture search (NAS) (Zoph & Le, 2017; Liu et al., 2019; Real et al., 2019) aims at automatically searching for high-performance architectures of neural networks to reduce humans' burden in manually designing them. The research of NAS has made remarkable progress in the past few years and achieved promising results. In existing NAS methods, model weights are learned by minimizing a training loss and the architecture is learned by minimizing an averaged validation loss (AVL). The AVL, calculated by taking the mean of validation losses computed on individual examples in a fixed validation set, reflects the average-case performance of a neural architecture. An architecture learned by minimizing the AVL focuses on achieving great average-case performance, but may have poor worst-case performance (Shu et al., 2020) (empirical justification is in Sec. 4). Consider the evaluation of a deep neural network based treatment recommendation system. Given a validation set where 99.99% of patients have common diseases and 0.01% of patients have rare diseases, a treatment recommendation system which can accurately recommend treatment plans for common diseases but performs poorly on rare diseases will have high average evaluation accuracy on this validation set. However, from the clinical perspective, such a system cannot be safely used in clinical practice since it cannot properly deal with patients with rare diseases (i.e., having poor worst-case performance).

To address the limitation of average-case evaluation, Shu et al. (2020) proposed adversarial examination, which dynamically selects a sequence of validation sets so that the performance of a model (with fixed weight parameters) evaluated on these validation sets decreases. By doing this, the model's worst-case performance (or weakness) can be identified. The notion of "adversarial" comes from the fact that the validation sets are selected to minimize the model's performance. While this work can evaluate the worst-case performance of a fixed model, it does not provide a mechanism to leverage the evaluation results to retrain the model for improving its worst-case performance.

We aim to bridge this gap and propose to generate adversarial validation examples to measure and improve the worst-case performance of a neural architecture. In our framework, there is a "learner" model and a "tester" model. The tester generates adversarial validation data using a deep generative model (Goodfellow et al., 2014a) to measure the worst-case performance of the learner. The learner updates its architecture by minimizing the loss on the generated adversarial validation data to improve its worst-case performance. Our method is based on multi-level optimization (Sato et al., 2021), which consists of four levels of nested optimization problems. At the first level, we train model weights of the learner with its architecture tentatively fixed. At the second level, a deep generative model (DGM) is trained. At the third level, an auxiliary model is trained using data generated by the DGM to verify the fidelity of generated data. At the fourth level, the DGM generates adversarial validation data; the learner updates its architecture by minimizing

the loss on the generated data; and the DGM updates its hyperparameters by minimizing the loss of the auxiliary model on a human-labeled validation set.

The major contributions of this paper are as follows:

- We propose a general framework to evaluate and improve the worst-case performance of neural architectures in NAS.

- We demonstrate the effectiveness of our method on various datasets.

## 2 Related work

**Neural architecture search (NAS).** Existing NAS methods can be roughly grouped into three categories: 1) reinforcement learning (RL) based methods; 2) evolutionary algorithm based methods; and 3) differentiable methods. In RL-based approaches (Zoph & Le, 2017; Pham et al., 2018; Zoph et al., 2018), a policy is learned to iteratively generate new architectures by maximizing a reward which is validation accuracy. Evolutionary algorithm based approaches (Liu et al., 2018b; Real et al., 2019) represent architectures as individuals in a population. Individuals with high fitness scores (validation accuracy) have the privilege to generate offspring, which replaces individuals with low fitness scores. Differentiable approaches (Cai et al., 2019; Liu et al., 2019; Xie et al., 2019) adopt a network pruning strategy. On top of an over-parameterized network, the importance weights of operators are learned using gradient descent. Operators with close-to-zero weights are pruned. Previous NAS methods cannot improve architectures' worst-case performance. Our work aims to bridge this gap.

**Adversarial learning.** Our formulation involves a mini-max optimization problem, which is analogous to that in adversarial learning (Goodfellow et al., 2014a) for data generation (Goodfellow et al., 2014a; Yu et al., 2017), domain adaptation (Ganin & Lempitsky, 2015), adversarial attack and defense (Goodfellow et al., 2014b), active learning (Mayer & Timofte, 2020), etc. Different from existing works, our work is featured with a tester which generates adversarial validation data to minimize the validation performance of a learner while the learner retrains itself to improve performance on generated validation data. Shu et al. (2020) proposed to use an adversarial examiner to identify the weakness of a trained model. Our work differs from (Shu et al., 2020) in that we continuously update a learner model based on how it performs on the validation examples that are generated dynamically by a tester model, while the learner model in (Shu et al., 2020) is fixed and not affected by examination results. Such et al. (2019) proposed to learn a generative adversarial network (Goodfellow et al., 2014a) to create synthetic examples which are used to train an NAS model. Our work differs from (Goodfellow et al., 2014a) in that we generate adversarial validation data to evaluate the worst-case performance of a model while Such et al. (2019) do not consider the notion of "adversarial".

**Curriculum learning (CL).** CL has been widely studied (Bengio et al., 2009; Kumar et al., 2010; Spitkovsky et al., 2010; Lee & Grauman, 2011; Jiang et al., 2014; Tsvetkov et al., 2016; Graves et al., 2017; Matiisen et al., 2019; Platanios et al., 2019), where a sequence of training datasets with increasing levels of difficulty are used for model training. Some CL works (Bengio et al., 2009; Spitkovsky et al., 2010; Tsvetkov et al., 2016; Platanios et al., 2019; Zhou et al., 2020d) use task-specific prior knowledge to measure hardness of examples, which are not generalizable across a variety of tasks. Self-paced CL (Kumar et al., 2010; Jiang et al., 2014; 2015; Zhou et al., 2021) automatically measures example hardness using training losses and the hardness can be dynamically adjusted. Our work differs from these previous works in that our work dynamically generates adversarial validation data for model evaluation while previous works select hard data examples for model training.

## 3 Method

### 3.1 Overview

In our framework, there is a learner model and a tester model, where the learner learns to perform a target task such as classification, regression, etc. The tester model uses a deep generative model (DGM) (Goodfellow et al., 2014a) to generate adversarial validation examples to evaluate the worst-case performance of the learner. These validation examples are generated in a way that the learner's performance on these examples (referred to as generative

adversarial validation examples (GAVEs)) decreases. The learner continuously retrains its model by maximizing the performance on GAVEs, to improve its worst-case performance. Note that GAVEs are different from adversarial examples (Goodfellow et al., 2014b) in the adversarial robustness literature. GAVEs are brand new examples which are generated by a DGM and have significant visual and semantic differences with real examples in a given dataset. In contrast, adversarial examples are created by adding small perturbations to real examples, and are very similar to real examples both visually and semantically. GAVEs focus on evaluating the worst-case performance of an ML model over the global landscape of a data distribution (which is the focus of our work) while adversarial examples focus on evaluating the robustness of an ML model around a local example. In Appendix B.3, we performed an evaluation of our method on adversarial examples.

Besides decreasing the learner's validation performance, generated validation examples should be "meaningful". It is possible that generated validation examples are of poor quality, which are outliers or have incorrect class labels. Using low quality validation examples to guide the learning of the learner may render the learner to overfit these examples. To address this problem, we evaluate the "meaningfulness" of generated examples by checking how useful they are when used to train an auxiliary model for performing the target task. If the auxiliary model trained on generated data achieves good validation performance on human-labeled real data examples, the generated examples are considered as being meaningful.

In our framework, both the learner and the tester perform learning. The learner aims to improve its worst-case performance. The tester learns to generate validation examples that can decrease the learner's validation performance and are meaningful. The learner has a learnable architecture. Since the learner and tester perform adversarial learning jointly, it is important to let them have similar model capacities to prevent one of them from being dominated by the other. Towards this goal, we make hyperparameters of the tester learnable. In Appendix B.2.2, we provided experiments which show that learning the hyperparameters of the tester yields better performance than not learning them.

### 3.2 A multi-level optimization based framework

In our framework, the learning of the learner and the tester is organized into four stages.

**Stage I.** Let $A$ denote the learner's architecture and $L_{tgt}(\cdot)$ denote the target task's training loss. At this stage, the learner trains its network weights $W$ on a training set $D^{(tr)}$:

$$W^*(A) = \text{argmin}_W \ L_{tgt}(W, A, D^{(tr)}). \tag{1}$$

The architecture $A$ is used to define the training loss, but it is not learned at this stage. If $A$ is learned by minimizing this training loss, a trivial solution will be yielded where the model can perfectly overfit the training data but will generalize poorly on test data. Let $W^*(A)$ denote the optimally learned $W$ at this stage. Note that $W^*$ depends on $A$ because $W^*$ depends on the training loss and the training loss is a function of $A$.

**Stage II.** At the second stage, the tester trains a data generation model which generates both input data and output labels. Let $G$ and $B$ denote weight parameters and hyperparameters of the data generation model. Let $L_{dg}$ denote a data generation loss. At this stage, we solve the following optimization problem:

$$G^*(B) = \text{argmin}_G \ L_{dg}(G, B, D^{(tr)}), \tag{2}$$

$G$ is trained on $D^{(tr)}$ by switching the input and output in each training example and $B$ is tentatively fixed.

**Stage III.** At the third stage, we apply the generator $G^*(B)$ trained at the second stage to generate a validation dataset $f(G^*(B))$. Then we train an auxiliary model $V$ on $f(G^*(B))$ to perform the target task. At this stage, we solve the following optimization problem:

$$V^*(G^*(B)) = \text{argmin}_V \ L_{tgt}(V, f(G^*(B))). \tag{3}$$

**Stage IV.** At the fourth stage, we evaluate the learner's model $W^*(A)$ on the validation set $f(G^*(B))$. For the tester, it aims to generate an adversarial validation set by maximizing the validation loss w.r.t $B$. For the learner, it aims to perform well under worst-case scenarios by minimizing the loss on the generative adversarial validation set w.r.t $A$.

On the other hand, we evaluate the auxiliary model $V^*(G^*(B))$ on a human-labeled validation set $D^{(val)}$. The tester aims to minimize this loss w.r.t $B$ to encourage the generated validation examples to be meaningful. When improving the worst-case performance of $A$, we would like $A$ to maintain its average-case performance as well. We evaluate the average-case loss of $A$ on $D^{(val)}$ and update $A$ by minimizing the average-case loss. At this stage, we solve the following optimization problem:

$$\min_A \max_B L_{tgt}(W^*(A), A, f(G^*(B))) - \lambda L_{tgt}(V^*(G^*(B)), D^{(val)}) + \gamma L_{tgt}(W^*(A), A, D^{(val)}), \tag{4}$$

where $\lambda$ and $\gamma$ are tradeoff parameters. $\lambda$ strikes the balance between the worst-case-ness and meaningfulness of generated data. $\gamma$ strikes the balance between worst-case and average-case performance.

**Multi-level optimization.** Putting these pieces together, we have the following multi-level optimization framework.

$$\begin{aligned}
\min_A \max_B \quad & L_{tgt}(W^*(A), A, f(G^*(B))) - \lambda L_{tgt}(V^*(G^*(B)), D^{(val)}) + \gamma L_{tgt}(W^*(A), A, D^{(val)}) \\
s.t. \quad & V^*(G^*(B)) = \text{argmin}_V \ L_{tgt}(V, f(G^*(B))) \\
& G^*(B) = \text{argmin}_G \ L_{dg}(G, B, D^{(tr)}) \\
& W^*(A) = \text{argmin}_W \ L_{tgt}(W, A, D^{(tr)})
\end{aligned} \tag{5}$$

### 3.3 Reduce search costs

Before running our method on a dataset $D$, we first pretrain the generative model and auxiliary model on the same dataset $D$. The total search cost includes pretraining the generative model and auxiliary model and running our multi-level optimization based framework to learn all variables in Eq.(5). We utilize the following approaches to reduce search and memory costs.

- The frequencies of calculating hypergradients of architecture $A$, generator $G$, and hyperparameters $B$ and updating these variables are reduced to every 8 iterations (i.e., mini-batches) instead of every mini-batch. We empirically found that by doing this, the computational costs were substantially reduced without significantly compromising image classification performance. For the rest of parameters, they were updated in every iteration as usual.

- On generated data in stage IV, we added a decorrelation regularizer (Cogswell et al., 2015), which accelerated convergence greatly and allowed us to reduce epoch number from 50 to 25 without compromising the quality of convergence.

- We adopted parameter tying to make the representation learning layers of the auxiliary model and those of the discriminator in FQ-GAN (Zhao et al., 2020) (which is used as the data generation model in our framework) share the same weight parameters. These parameters account for more than 95% of each model's total parameters. Tying these parameters greatly reduces parameter number, thereby decreasing computational costs of training them.

- We reduced the pretraining time of the FQ-GAN to 4 hours on either CIFAR-10 or CIFAR-100, by 1) using the methods proposed in (Sinha et al., 2020a;b); 2) reducing the number of iterations by half.

- For pretraining the auxiliary model, the weight parameters of its feature extraction layers are set to those of the pretrained discriminator in FQ-GAN. We only need to pretrain the light-weight classification head (two feedforward layers) of the auxiliary model, which was efficiently finished in 20 minutes.

In addition, we provide an alternative way of generating adversarial validation examples, by automatically learning a data augmentation policy to augment data from original data and using augmented examples as adversarial validation data. The number of parameters in a data augmentation policy is much smaller than that in a deep generative model, and therefore is computationally more efficient to train. Following (Li et al., 2020), we learn a differentiable augmentation policy $P$ containing a set of sub-policies where each sub-policy consists of two operations and each operation has a probability and magnitude. The goal is to learn how to select sub-policies, which is formulated as a differentiable optimization problem using the Gumbel-Softmax (Jang et al., 2016) method. Given a training set $D^{(tr)}$, we randomly

select a subset of $D^{(tr)}$. Then the learned $P$ is applied to the selected subset to generate a set of augmented examples $f(P, D^{(tr)})$.

The formulation in Eq.(5) becomes

$$
\begin{aligned}
&\min_A \max_P \quad L_{tgt}(W^*(A), A, f(P, D^{(tr)})) - \lambda L_{tgt}(V^*(P), D^{(val)}) + \gamma L_{tgt}(W^*(A), A, D^{(val)}) \\
&s.t. \quad V^*(P) = \text{argmin}_V \ L_{tgt}(V, f(P, D^{(tr)})) \\
&\qquad\quad W^*(A) = \text{argmin}_W \ L_{tgt}(W, A, D^{(tr)})
\end{aligned}
\tag{6}
$$

## 4 Experiments

We apply our method for neural architecture search in image classification. We use a conditional generative adversarial network (GAN) (Goodfellow et al., 2014a; Mirza & Osindero, 2014) – FQ-GAN (Zhao et al., 2020) – to generate validation examples. We randomly sample a class label $y$ and a random noise vector $z$, then feed $y$ and $z$ into FQ-GAN to generate an image $x$. $(x, y)$ is regarded as a generated image-label pair. We treat the momentum decay $\lambda$ and FQ weight $\alpha$ in FQ-GAN as the learnable hyperparameters $B$ in the data generation model. They are continuous scalars, which can be efficiently optimized using gradient-based methods by maximizing the loss in Eq.(4). We initialize them to the values reported in (Zhao et al., 2020). Please refer to Appendix B.2.2 for more details regarding how these hyperparameters influence data generation. Following (Liu et al., 2019), we first perform architecture search which finds an optimal cell, then perform architecture evaluation which composes multiple copies of the searched cell into a large network, trains it from scratch, and evaluates it on a test set. Please refer to the appendix for detailed hyperparameter settings and additional results, such as computational costs, evaluation on adversarial examples, etc.

### 4.1 Datasets

We used five datasets: CIFAR-100 (Krizhevsky et al., 2009), CIFAR-10 (Krizhevsky & Hinton, 2010), ImageNet (Deng et al., 2009), ImageNet-C (Hendrycks & Dietterich, 2019), and CIFAR-10-C (Hendrycks & Dietterich, 2019). CIFAR-100 and CIFAR-10 contain 50K training images and 10K testing images, from 100 and 10 classes respectively. For each dataset, we split the original 50K training set into a 25K new training set and a 25K validation set. ImageNet contains a training set of 1.3M images and a test set of 50K images, from 1000 object classes. Following (Xu et al., 2020), 10% of the 1.3M training images are randomly sampled to form a new training set and another 2.5% of the 1.3M training images are randomly sampled to form a new architecture validation set. The ImageNet-C dataset consists of 15 diverse corruption types applied to validation images of ImageNet. The corruptions are drawn from four main categories - noise, blur, weather, and digital. CIFAR-10-C is a dataset generated by adding 15 common corruptions and 4 extra corruptions to CIFAR-10 test images.

Architectures are searched on CIFAR-100, CIFAR-10, and ImageNet, which are the most broadly used benchmark datasets in the NAS literature. To evaluate the worst-case performance of searched architectures, we manually select 2000 (20 per class) "worst-case" examples (whose class labels are considered by humans as being challenging to recognize) from the test set of CIFAR-100. In addition, we test these architectures on CIFAR-10-C and ImageNet-C.

### 4.2 Experimental settings

Our framework is a general one that can be used together with any differentiable search method. Specifically, we apply our framework to the following NAS methods: DARTS (Liu et al., 2019), P-DARTS (Chen et al., 2019), PC-DARTS (Xu et al., 2020), and PR-DARTS (Zhou et al., 2020b). The formulation in Eq.(5) is applied to DARTS and the formulation in Eq.(6) is applied to P-DARTS, PC-DARTS, and PR-DARTS.

During architecture evaluation, the combination of the training data and validation data is used to train a large network stacking multiple copies of the searched cell. In addition to searching for architectures directly on ImageNet data, following (Liu et al., 2019), we also evaluate the architectures searched using CIFAR-10 and CIFAR-100 on ImageNet: given a cell searched using CIFAR-10 and CIFAR-100, multiple copies of it compose a large network, which is then trained on the 1.3M training data of ImageNet and evaluated on the 50K testing data.

The auxiliary model is set to ResNet-18 (He et al., 2016b). The tradeoff parameters $\lambda$ and $\gamma$ are tuned using a 5k held-out dataset in $\{0.1, 0.5, 1, 2, 3\}$. In most experiments, $\lambda$ is set to 1 except for P-DARTS and PC-DARTS. For

Table 1: Classification error on worst-case, average-case, and all examples in CIFAR-100 test set.

| Method | Worst-case | Average-case | All |
|---|---|---|---|
| Darts2nd | 30.07±0.36 | 18.82±0.39 | 20.58±0.44 |
| SPCL-darts2nd | 31.82±0.12 | 17.60±0.24 | 20.14±0.27 |
| DIHCL-darts2nd | 30.88±0.25 | 17.42±0.33 | 19.86±0.42 |
| Ours-darts2nd (ours) | **27.01**±0.17 | **16.08**±0.13 | **18.55**±0.09 |
| Pdarts | 27.93±0.36 | 15.82±0.29 | 17.96±0.15 |
| SPCL-pdarts | 28.08±0.25 | 15.61±0.11 | 17.71±0.17 |
| DIHCL-pdarts | 28.36±0.28 | 15.09±0.26 | 18.05±0.28 |
| Ours-pdarts (ours) | **25.51**±0.16 | **14.12**±0.15 | **17.10**±0.13 |
| Pcdarts | 28.42±0.15 | 14.95±0.11 | 17.43±0.13 |
| SPCL-pcdarts | 27.06±0.28 | 14.26±0.17 | 17.25±0.09 |
| DIHCL-pcdarts | 27.94±0.22 | 15.70±0.19 | 17.69±0.15 |
| Ours-pcdarts (ours) | **24.92**±0.09 | **13.36**±0.11 | **16.17**±0.08 |
| Prdarts | 25.17±0.14 | 14.77±0.18 | 16.48±0.06 |
| SPCL-prdarts | 25.33±0.17 | 14.45±0.14 | 16.76±0.10 |
| DIHCL-prdarts | 25.80±0.31 | 14.95±0.26 | 16.83±0.14 |
| Ours-prdarts (ours) | **23.02**±0.11 | **14.82**±0.10 | **16.13**±0.02 |

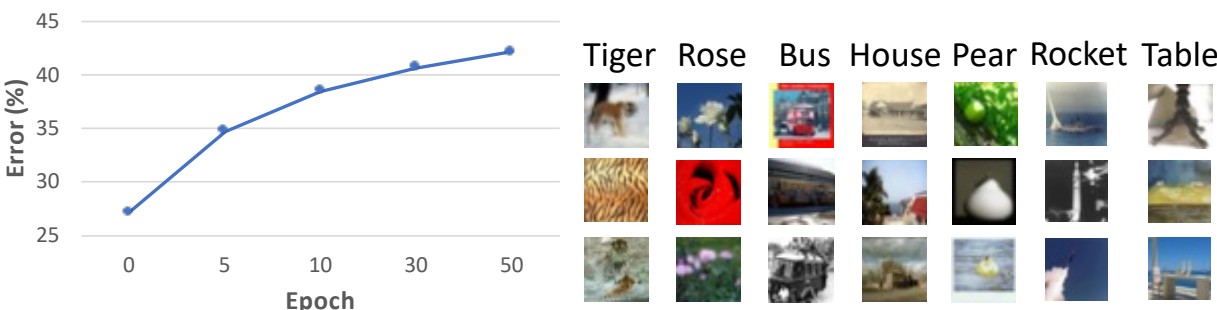

Figure 1: (Left) Errors of a pretrained ResNet-18 on validation sets generated at different epochs. (Right) Randomly-sampled images from CIFAR-100 test set where Ours-darts2nd makes correct predictions while vanilla Darts-2nd makes incorrect predictions.

P-DARTS, $\lambda$ is set to $0.5$ for CIFAR-10 and $1$ for CIFAR-100. For PC-DARTS, we use $\lambda = 3$ and $\lambda = 0.1$ for CIFAR-10 and CIFAR-100, respectively. $\gamma$ is set to $1$. Before running our method on a dataset $D$, we pretrain the data generation model (FQ-GAN) and auxiliary model (ResNet-18) on $D$. Note that our method did not unfairly use more data than baselines. The experiments were conducted on Nvidia 1080Ti GPUs.

We compare with the following curriculum learning (CL) methods: 1) self-paced CL (SPCL) (Jiang et al., 2015), and 2) dynamic instance hardness guided CL (DIHCL) (Zhou et al., 2020c). These methods were originally designed for selecting training examples that have large training losses. We adapt them to select validation examples that have large validation losses, which are used to update architecture parameters.

## 4.3 Results on "worst-case" test examples

**Results on worst-case examples in CIFAR-100 test set.** Table 1 shows classification errors on 2000 "worst-case" examples selected from the CIFAR-100 test set. From this table, we can see that when our method is applied to different NAS baselines including DARTS2nd, PC-DARTS, P-DARTS, and PR-DARTS, the classification errors of these baselines on worst-case test examples are significantly reduced (please refer to Appendix B.7 for statistical significance test results). This demonstrates the effectiveness of our framework in improving the worst-case performance of NAS.

|  | Epoch 0 | Epoch 5 | Epoch 10 | Epoch 30 | Epoch 50 |
| Bridge | | | | | |
| Mushroom | | | | | |
| Lamp | | | | | |

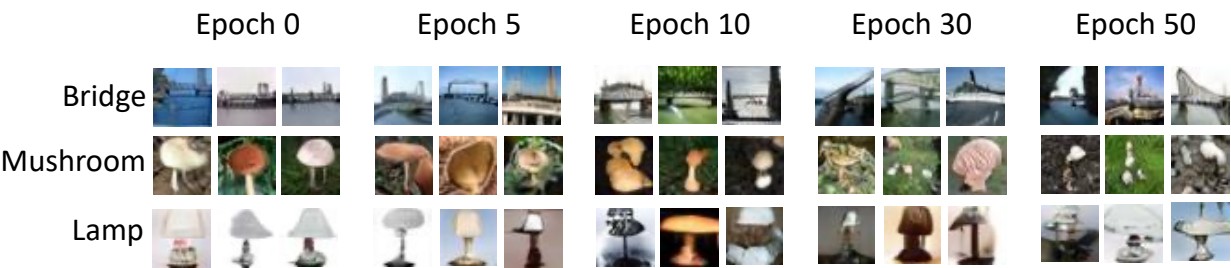

Figure 2: Randomly sampled images from validation sets created at different epochs.

In our method, the learner improves its architecture's worst-case performance by minimizing the loss on the adversarial validation data generated by the tester. The adversarial validation sets are generated in a way that the learner's performance on these sets decreases. To verify this, we apply an ResNet-18 pretrained on the training set to make predictions on validation sets generated at different epochs. Figure 1(left) shows that the errors consistently increase with epoch number. Figure 2 shows some randomly sampled validation examples generated at different epochs. As can be seen, as the epoch increases, the images are more and more difficult to recognize (we evaluated the quality of generated validation examples in Appendix B.6).

These generative adversarial validation sets can help the learner to identify the worst-case weakness of its architecture and provide guidance on how to improve it. Different from baseline NAS methods which optimize architectures by minimizing an average-case validation loss calculated on a single fixed validation set, our method improves its architecture by minimizing the loss on each generated adversarial validation set. By doing this, architectures searched by our method can make more accurate predictions on worst-case examples during test time.

In this table, we also see that our method works better than the two curriculum learning (CL) methods: SPCL and DIHCL. In our method, adversarial validation data generation and architecture search are performed jointly in an end-to-end framework whereas in the two baselines, validation example selection is performed separately from architecture search. Performing the two tasks end-to-end enables the generation of adversarial validation examples to be guided dynamically by the quality of the learner's architectures.

In Table 1, we also report performance on all test images and the 8000 "average-case" images. As can be seen, our method's performance on average-case test data is on par with that of baselines. This demonstrates that our framework is able to improve NAS' worst-case performance without sacrificing average-case performance. The reason is that in Eq.(4), the architecture is updated by simultaneously minimizing the worse-case validation loss (the first term) and the average-case validation loss (the second term). By seeking a proper trade-off between these two losses, our framework is able to maintain average-case performance while significantly boosting worst-case performance.

Table 2: Average classification error (%) on CIFAR-10-C. Architectures are searched on CIFAR-10.

| Method | Error |
|---|---|
| Darts2nd | 23.8±0.7 |
| SPCL-Darts2nd | 23.5±0.9 |
| DIHCL-Darts2nd | 22.9±1.2 |
| Ours-Darts2nd | **20.1**±0.4 |
| Pdarts | 22.4±1.1 |
| SPCL-Pdarts | 22.7±0.7 |
| DIHCL-Pdarts | 22.1±1.0 |
| Ours-Pdarts | **18.9**±0.9 |
| Pcdarts | 23.0±0.6 |
| SPCL-Pcdarts | 22.8±0.4 |
| DIHCL-Pcdarts | 22.5±0.9 |
| Ours-Pcdarts | **19.8**±0.5 |
| Prdarts | 21.5±0.9 |
| SPCL-Prdarts | 21.2±0.6 |
| DIHCL-Prdarts | 21.8±1.0 |
| Ours-Prdarts | **19.1**±0.3 |

Table 3: Mean corruption error (mCE, %) on ImageNet-C. Architectures are searched on ImageNet.

| Method | mCE |
|---|---|
| Pcdarts | 78.8 |
| SPCL-Pcdarts | 78.4 |
| DIHCL-Pcdarts | 79.1 |
| Ours-Pcdarts | **75.4** |

**Results on CIFAR-10-C and ImageNet-C.** We evaluate different methods on CIFAR-10-C and ImageNet-C (Hendrycks & Dietterich, 2018). For CIFAR-10-C experiments, the architectures were searched on CIFAR-10. For ImageNet-C experiments, the architectures were searched on ImageNet and our framework was applied to PC-DARTS (we did not experiment with other NAS baselines such as DARTS which are computationally too costly to perform architecture search on ImageNet). Table 2 and 3 shows the average classification error on CIFAR-10-C and

Table 4: Test errors on CIFAR-10 and CIFAR-100 in the evaluation of robustness against performance collapse. † denotes that the initial channel number is 16 and cell number is 8. ‡ denotes that the initial channel number is 36 and cell number is 20. We compared with DARTS (Liu et al., 2019), RDARTS-L2 (Zela et al., 2019), DARTS-ES (Zela et al., 2019), DARTS- (Chu et al., 2020a), SDART (Chen & Hsieh, 2020b), and SPCL (Jiang et al., 2015).

| Data | Space | DARTS | RDARTS-L2 | DARTS-ES | DARTS- | SDART | SPCL | Ours |
|------|-------|-------|-----------|----------|--------|-------|------|------|
| CIFAR-10 | S1† | 4.69 | 3.46 | 3.93 | 3.34 | 3.26 | 4.02 | **3.08** |
| | S1‡ | 3.84 | 2.78 | 3.01 | 2.68 | 2.73 | 3.29 | **2.55** |
| | S2 | 5.54 | 3.31 | 4.07 | 4.03 | 3.11 | 4.50 | **2.91** |
| | S3 | 3.92 | 2.51 | 3.55 | 2.95 | 3.07 | 4.26 | **2.46** |
| | S4 | 8.33 | 3.56 | 4.69 | 4.14 | 3.49 | 5.72 | **3.28** |
| CIFAR-100 | S1 | 29.46 | 24.25 | 28.37 | 22.41 | 22.33 | 26.72 | **21.38** |
| | S2 | 26.05 | 22.24 | 23.25 | 21.61 | 20.56 | 24.10 | **19.45** |
| | S3 | 28.90 | 23.99 | 23.73 | 21.13 | 21.08 | 20.79 | **19.92** |
| | S4 | 22.85 | 21.94 | 21.26 | 21.55 | 21.25 | 23.38 | **20.33** |

the mean Corruption Error on ImageNet-C (Hendrycks & Dietterich, 2018), respectively. Lower is better. Our methods outperform baselines, which demonstrates that our methods are more robust than baselines against data corruptions in CIFAR-10-C and ImageNet-C. The reason is: examples in CIFAR-10-C and ImageNet-C can be considered as a type of worst-case examples; our method measures and robustifies models' worst-case performance by using deep generative models to generate worst-case examples, and therefore has better capability to handle examples in CIFAR-10-C and ImageNet-C.

## 4.4 Robustness against performance collapse

Next, we present another reason for our method's effectiveness. Our method optimizes the architecture by minimizing the loss on each generated adversarial validation set. As a result, the learned architecture is more robust against performance collapse. Several studies (Zela et al., 2019; Chu et al., 2020a; Chen & Hsieh, 2020b) have shown that differentiable NAS methods are prone to performance collapse. To empirically show the robustness of our method, we evaluate it on four search spaces designed for measuring robustness in (Zela et al., 2019). Following RobustDARTS-L2 (Zela et al., 2019), we set cell number to 8 and initial channel number to 16 for both CIFAR-10 and CIFAR-100. The same as the protocols in DARTS (Liu et al., 2019) and RobustDARTS (Zela et al., 2019), architecture search runs for 4 times with random initialization.

A searched architecture is retrained from scratch for a few epochs. The architecture achieving the highest validation accuracy is selected. Architecture evaluation is performed on selected architectures. Table 4 shows the results. As can be seen, on the four spaces, our method outperforms baseline methods. This demonstrates that our method is more robust against performance collapse.

## 4.5 Overall errors on CIFAR-10 and ImageNet

In this section, we evaluate architectures' overall performance on entire test sets, which include both difficult examples and average-case examples.

**Results on CIFAR-10.** Table 5 shows the classification error (%), number of weight parameters (millions), and search cost (GPU days) of different NAS methods on the entire test set of CIFAR-10. When our framework is applied to NAS baselines, the overall performance of these baselines is significantly improved. This is because our framework can improve worst-case performance without sacrificing average-case performance, as shown in Table 1, which results in improved overall performance.

**Results on ImageNet.** Table 23 shows results on ImageNet. Ours-pcdarts-imagenet where the architecture is searched on ImageNet performs much better than Pcdarts-imagenet and achieves the lowest errors among all methods in Table 23. Ours-pdarts-cifar100, Ours-pdarts-cifar10, and Ours-darts2nd-cifar10, where the architectures are searched on CIFAR-10 or CIFAR-100 and evaluated on ImageNet, outperform their corresponding baselines Pdarts-

Table 5: Results on CIFAR-10, including classification error (%) on the test set, number of parameters (millions) in the searched architecture, and search cost (GPU days). Ours-darts2nd denotes that our method is applied to the search space of DARTS. Similar meanings hold for other notations in such a format. * means the results are taken from DARTS$^-$ (Chu et al., 2020a), $\beta$-DARTS (Ye et al., 2022), and AGNAS (Sun et al., 2022). † means we re-ran this method for 10 times. The search cost is measured by GPU days.

| Method | Error | Param. | Cost |
|---|---|---|---|
| *ResNet (He et al., 2016a) | 6.43 | 1.7 | - |
| *DenseNet (Huang et al., 2017) | 3.46 | 25.6 | - |
| *PNAS (Liu et al., 2018a) | 3.41±0.09 | 3.2 | 150 |
| *ENAS (Pham et al., 2018) | 2.89 | 4.6 | 0.5 |
| *AmoebaNet (Real et al., 2019) | 2.55±0.05 | 3.1 | 3150 |
| *GDAS (Dong & Yang, 2019) | 2.93 | 3.4 | 0.2 |
| *R-DARTS (Zela et al., 2020) | 2.95±0.21 | - | 1.6 |
| *DARTS+PT (Wang et al., 2021) | 2.61±0.08 | 3.0 | 0.8 |
| *DARTS$^-$ (Chu et al., 2020a) | 2.59±0.08 | 3.3 | 0.4 |
| *DropNAS (Hong et al., 2020) | 2.58±0.14 | 4.4 | 0.7 |
| *DrNAS (Chen et al., 2020) | 2.54±0.03 | 4.0 | 0.4 |
| *ISTA-NAS (Yang et al., 2020) | 2.54±0.05 | 3.3 | 0.1 |
| *MiLeNAS (He et al., 2020) | 2.51±0.11 | 3.9 | 0.3 |
| *GAEA (Li et al., 2021) | 2.50±0.06 | - | 0.1 |
| *PDARTS-ADV (Chen & Hsieh, 2020a) | 2.48±0.02 | 3.4 | 1.1 |
| *DOTS (Gu et al., 2021) | 2.49±0.06 | 4.1 | 0.3 |
| *$\beta$-DARTS (Ye et al., 2022) | 2.53±0.08 | 3.8 | 0.4 |
| *AGNAS (Sun et al., 2022) | 2.53±.003 | 3.6 | 0.4 |
| *RF-DARTS (Zhang et al., 2023) | 2.60 | 4.6 | - |
| *Darts2nd (Liu et al., 2019) | 2.76±0.09 | 3.1 | 4.0 |
| SPCL-darts2nd (Jiang et al., 2015) | 2.81±0.23 | 3.2 | 5.1 |
| DIHCL-darts2nd (Zhou et al., 2020c) | 2.86±0.10 | 3.4 | 5.4 |
| Ours-darts2nd (ours) | **2.68**±0.03 | 3.1 | 4.0 |
| †Pcdarts (Xu et al., 2020) | 2.57±0.07 | 3.9 | 0.1 |
| SPCL-pcdarts (Jiang et al., 2015) | 2.69±0.14 | 3.9 | 0.2 |
| DIHCL-pcdarts (Zhou et al., 2020c) | 2.73±0.05 | 4.0 | 0.4 |
| Ours-pcdarts (ours) | **2.50**±0.05 | 3.7 | 0.1 |
| *Pdarts (Chen et al., 2019) | 2.54±0.04 | 3.6 | 0.3 |
| SPCL-pdarts (Jiang et al., 2015) | 2.77±0.25 | 3.8 | 0.6 |
| DIHCL-pdarts (Zhou et al., 2020c) | 2.85±0.12 | 3.8 | 0.9 |
| Ours-pdarts (ours) | **2.48**±0.04 | 3.6 | 0.3 |
| †Prdarts (Zhou et al., 2020b) | 2.37±0.03 | 3.4 | 0.2 |
| SPCL-prdarts (Jiang et al., 2015) | 2.44±0.05 | 3.3 | 0.3 |
| DIHCL-prdarts (Zhou et al., 2020c) | 2.48±0.06 | 3.5 | 0.5 |
| Ours-prdarts (ours) | **2.31**±0.02 | 3.3 | 0.2 |

cifar100, Pdarts-cifar10, and Darts2nd-cifar10. These results further demonstrate the effectiveness of our method in improving NAS' overall performance.

## 4.6 Ablation studies

To investigate the importance of encouraging the generated validation examples to be meaningful, we perform an ablation study of "adversarial only": the tester generates validation examples solely by maximizing the learner's validation loss, without considering their meaningfulness. Accordingly, the third stage in our framework where the tester trains an auxiliary model on generated validation examples is removed and $\lambda$ at the fourth stage is set to 0. For

Table 6: Results on ImageNet, including top-1 and top-5 classification errors on the test set. * means the results are taken from DARTS$^-$ (Chu et al., 2020a), DrNAS (Chen et al., 2020), and $\beta$-DARTS (Ye et al., 2022). The rest of the notations are the same as those in Table 5.

| Method | Top-1 | Top-5 |
|---|---|---|
| *Inception-v1 (Szegedy et al., 2015) | 30.2 | 10.1 |
| *MobileNet (Howard et al., 2017) | 29.4 | 10.5 |
| *ShuffleNet 2× (v2) (Ma et al., 2018) | 25.1 | 7.6 |
| *NASNet-A (Zoph et al., 2018) | 26.0 | 8.4 |
| *AmoebaNet-C (Real et al., 2019) | 24.3 | 7.6 |
| *SNAS-CIFAR10 (Xie et al., 2019) | 27.3 | 9.2 |
| *DSNAS-ImageNet (Hu et al., 2020) | 25.7 | 8.1 |
| *PCDARTS-CIFAR10 (Xu et al., 2020) | 25.1 | 7.8 |
| *ProxylessNAS-ImageNet (Cai et al., 2019) | 24.9 | 7.5 |
| *FairDARTS-ImageNet (Chu et al., 2019) | 24.4 | 7.4 |
| *DrNAS-ImageNet (Chen et al., 2020) | 24.2 | 7.3 |
| *PR-DARTS (Zhou et al., 2020b) | 24.1 | 7.3 |
| *DARTS$^-$-ImageNet (Chu et al., 2020a) | 23.8 | 7.0 |
| *DOTS (Gu et al., 2021) | 24.3 | 7.4 |
| *$\beta$-DARTS (Ye et al., 2022) | 23.9 | 7.0 |
| *RF-PCDARTS (Zhang et al., 2023) | 23.9 | 7.1 |
| *Darts2nd-cifar10 (Liu et al., 2019) | 26.7 | 8.7 |
| SPCL-darts2nd-cifar10 (Jiang et al., 2015) | 26.4 | 8.5 |
| DIHCL-darts2nd-cifar10 (Zhou et al., 2020c) | 26.8 | 8.9 |
| Ours-darts2nd-cifar10 (ours) | **25.8** | **8.2** |
| *Pdarts-cifar10 (Chen et al., 2019) | 24.4 | 7.4 |
| SPCL-pdarts-cifar10 (Jiang et al., 2015) | 24.4 | 7.4 |
| DIHCL-pdarts-cifar10 (Zhou et al., 2020c) | 24.6 | 7.5 |
| Ours-pdarts-cifar10 (ours) | **24.1** | **7.1** |
| *Pdarts-cifar100 (Chen et al., 2019) | 24.7 | 7.5 |
| SPCL-pdarts-cifar100 (Jiang et al., 2015) | 24.5 | 7.4 |
| DIHCL-pdarts-cifar100 (Zhou et al., 2020c) | 24.7 | 7.6 |
| Ours-pdarts-cifar100 (ours) | **24.0** | **7.1** |
| *Pcdarts-imagenet (Xu et al., 2020) | 24.2 | 7.3 |
| SPCL-pcdarts-imagenet (Jiang et al., 2015) | 24.0 | 7.2 |
| DIHCL-pcdarts-imagenet (Zhou et al., 2020c) | 24.1 | 7.2 |
| Ours-pcdarts-imagenet (ours) | **23.5** | **6.7** |

Table 7: Errors for "adversarial only" on test sets of CIFAR-100 (C100) and CIFAR-10 (C10).

| Method | Error (%) |
|---|---|
| Adversarial only (Darts2nd, C100) | 20.52±0.11 |
| Ours (Darts2nd, C100) | **18.55**±0.09 |
| Adversarial only (Pdarts, C100) | 18.45±0.16 |
| Ours (Pdarts, C100) | **16.17**±0.08 |
| Adversarial only (Darts2nd, C10) | 2.84±0.06 |
| Ours (Darts2nd, C10) | **2.68**±0.03 |

CIFAR-100, our method is applied to P-DARTS and DARTS-2nd. For CIFAR-10, our method is applied to DARTS-2nd. Table 7 shows the results. On both CIFAR-10 and CIFAR-100, it is more advantageous to generate validation examples that are both meaningful and result in the degradation of the learner's performance, rather than focusing solely on impairing the learner's performance with the generated validation examples. The reason is that without being constrained to be meaningful, the generated validation examples could be outliers that hurt NAS performance.

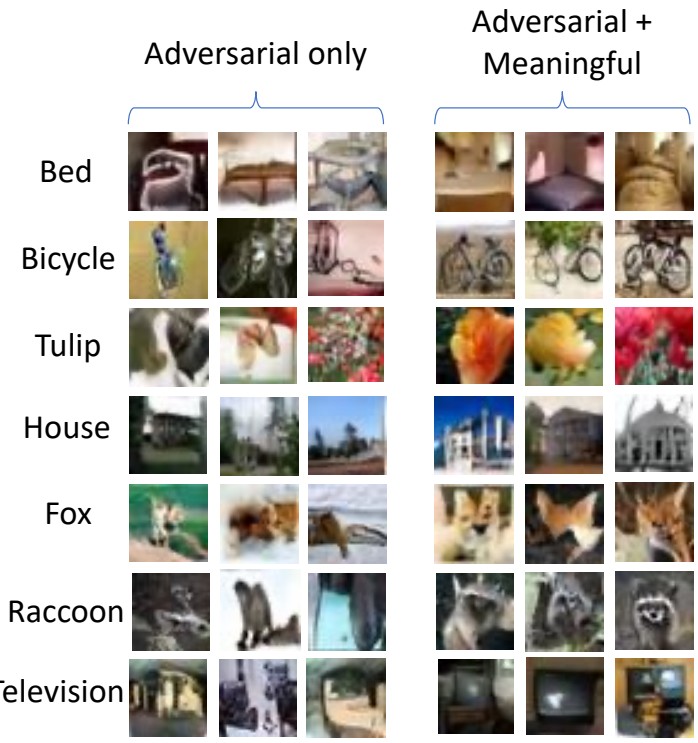

Figure 3: Randomly sampled images that are generated under "adversarial-only" and under "adversarial+meaningful".

Table 8: Errors on CIFAR-100 test set, under different tester networks.

| Method | Error(%) |
|---|---|
| SNGAN-darts2nd | 19.82±0.11 |
| BigGAN-darts2nd | 19.69±0.18 |
| FQGAN-darts2nd | **18.55**±0.09 |
| SNGAN-pcdarts | 17.84±0.09 |
| BigGAN-pcdarts | 17.91±0.07 |
| FQGAN-pcdarts | **17.10**±0.13 |
| SNGAN-pdarts | 17.22±0.09 |
| BigGAN-pdarts | 17.03±0.12 |
| FQGAN-pdarts | **16.17**±0.08 |

Figure 3 shows some images randomly sampled from validation examples generated under "adversarial only". As can be seen, these images are difficult to recognize even for humans or contain labeling errors. Even a highly accurate model cannot achieve good performance on such erratic examples. In contrast, validation images generated by "adversarial+meaningful" are more semantically meaningful. Our method promotes meaningfulness of generated examples by encouraging these examples to be useful for training a high-performance auxiliary model. The results demonstrate that this is an effective way of improving meaningfulness.

We perform another ablation study on how the data generation power of testers affects the learner's performance. We experimented with three testers: SN-GAN (Miyato et al., 2018), BigGAN (Brock et al., 2018), and FQ-GAN (Zhao et al., 2020). As reported in (Zhao et al., 2020), FQ-GAN performs better than BigGAN and SN-GAN in generating high-fidelity images. Table 8 shows that using FQ-GAN yields better NAS performance than using BigGAN and SN-GAN. The reason is: with a more powerful data generator, the tester can more effectively generate adversarial and

Table 9: Search cost (SC) in GPU hours and memory costs (MC) in MiB of different methods.

| Method | Error-C100 | Error-C10 | Param. | SC | MC |
|---|---|---|---|---|---|
| Darts2nd | 20.58±0.44 | 2.76±0.09 | 3.1 | 4.0 | 11053 |
| NCR-darts2nd (ours) | 18.42±0.11 | 2.67±0.03 | 3.3 | 6.1 | 19226 |
| CR-darts2nd (ours) | 18.55 ±0.09 | 2.68 ±0.03 | 3.1 | 3.9 | 10982 |
| Pcdarts | 17.96±0.15 | 2.57±0.07 | 3.9 | 0.1 | 10058 |
| NCR-pcdarts (ours) | 17.01±0.18 | 2.50±0.07 | 4.0 | 0.7 | 18429 |
| CR-pcdarts (ours) | 17.10±0.13 | 2.50±0.05 | 3.7 | 0.1 | 10071 |
| Pdarts | 17.43±0.13 | 2.54±0.04 | 3.6 | 0.3 | 9659 |
| NCR-pdarts (ours) | 16.14±0.06 | 2.48 ±0.05 | 3.8 | 1.1 | 18581 |
| CR-pdarts (ours) | 16.17±0.08 | 2.48±0.04 | 3.6 | 0.3 | 9597 |
| Prdarts | 16.48±0.06 | 2.37±0.03 | 3.4 | 0.2 | 10159 |
| NCR-prdarts (ours) | 16.14±0.02 | 2.31±0.01 | 3.4 | 0.8 | 20188 |
| CR-prdarts (ours) | 16.13±0.02 | 2.31±0.02 | 3.3 | 0.2 | 10195 |

meaningful validation examples. Validation examples with better quality can evaluate the learner more effectively and help to improve the learner's architecture.

In the appendix, we experimented additional ablation studies: 1) sensitivity analysis of $\lambda$ and $\gamma$; 2) do not learn the tester's hyperparameters, only learn its weight parameters; and 3) use a pretrained deep generative model as the tester, without learning the weight parameters or meta parameters of the tester in our framework. Our full method outperforms these ablation settings, which further demonstrates the effectiveness of the individual components in our method.

## 4.7 Search cost and memory cost

Table 9 shows search costs (GPU hours) and memory costs (MiB) of different methods. CR and NCR denote our method after and before applying the search cost reduction methods described in Section 3.3, respectively. As can be seen, the search costs of CR are no more than those of baselines, while the classification errors of CR are significantly lower than those of baselines. CR has lower costs than NCR while achieving classification performance similar to that of NCR. These observations demonstrate the effectiveness of these cost reduction methods. We applied some of these cost-saving strategies on baselines like DARTS, including decreasing the frequency of architecture updates from every iteration to every eighth iteration, and reducing the number of epochs from 50 to 25. However, these modifications resulted in a significant decline in performance. In light of that, we retained the default hyperparameters for the NAS baselines. Other cost-reduction methods, such as parameter tying and minimizing the pretraining time of the data generation model, cannot be applied to the NAS baselines.

# 5 Conclusions

We propose a multi-level optimization based framework where a tester model generates adversarial validation examples to measure the worst-case performance of a learner model. A learner model improves the worst-case performance of its architecture by minimizing the loss on the generated adversarial validation data. Experiments on various datasets demonstrate the effectiveness of our method.

## Broader Impact Statement

One potential negative societal impact of our work is: if the generated validation examples have low-fidelity, architectures searched based on these examples may make unexpected errors, which makes decision-making in mission-critical areas such as healthcare and finance unreliable. One major limitation of this work is that it is difficult to be applied to non-differentiable NAS methods, including those based on reinforcement learning and evolutionary algorithms.

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

## A  Full description of optimization algorithm

We use a well-established algorithm developed in (Liu et al., 2019) to solve the proposed four-level optimization problem. Theoretic convergence of this algorithm has been broadly analyzed in (Ghadimi & Wang, 2018; Grazzi et al., 2020; Ji et al., 2021; Liu et al., 2021; Yang et al., 2021). At each level of optimization problem, the optimal solution (on the left-hand side of the equal sign, marked with $^*$), its exact value is computationally expensive to compute. To address this problem, following (Liu et al., 2019), we approximate the optimal solution using a one-step gradient descent update and plug the approximation into the next level of optimization problem. In the sequel, we use $\nabla^2_{Y,X} f(X,Y)$ to denote $\frac{\partial f(X,Y)}{\partial X \partial Y}$. $\frac{\partial \cdot}{\partial \cdot}$ denotes partial derivative. $\frac{d \cdot}{d \cdot}$ denotes an ordinary derivative.

Following (Liu et al., 2019), we approximate $W^*(A)$ using one-step gradient descent update of $W$ w.r.t $L_{tgt}(W, A, D^{(tr)})$:

$$W^*(A) \approx W' = W - \eta_w \nabla_W L_{tgt}(W, A, D^{(tr)}). \tag{7}$$

Similarly, we approximate $G^*(B)$ using one-step gradient descent update of $G$ w.r.t $L_{dg}(G, B, D^{(tr)})$:

$$G^*(B) \approx G' = G - \eta_g \nabla_G L_{dg}(G, B, D^{(tr)}). \tag{8}$$

Plugging $G^*(B) \approx G'$ into $L_{tgt}(V, f(G^*(B)))$, we get an approximated objective. Then we approximate $V^*(G^*(B))$ using one-step gradient descent update of $V$ w.r.t the approximated objective:

$$V^*(G^*(B)) \approx V' = V - \eta_v \nabla_V L_{tgt}(V, f(G')). \tag{9}$$

Finally, we plug $W^*(A) \approx W'$ and $V^*(G^*(B)) \approx V'$ into $L_{tgt}(W^*(A), A, f(G^*(B))) - \lambda L_{tgt}(V^*(G^*(B)), D^{(val)}) + \gamma L_{tgt}(W^*(A), A, D^{(val)})$ and get an approximate objective. We update $A$ using gradient descent w.r.t the approximated objective:

$$A \leftarrow A - \eta_a \nabla_A L_{tgt}(W', A, f(G')) + \gamma L_{tgt}(W', A, D^{(val)}), \tag{10}$$

and update $B$ using gradient ascent:

$$B \leftarrow B + \eta_b \nabla_B (L_{tgt}(W', A, f(G')) - \lambda L_{tgt}(V', D^{(val)})). \tag{11}$$

$\nabla_A L_{tgt}(W', A, f(G'))$ can be computed as:

$$\nabla_A L_{tgt}(W', A, f(G')) \quad = \frac{dW'}{dA} \nabla_{W'} L_{tgt}(W', A, f(G')) + \nabla_A L_{tgt}(W', A, f(G')) \tag{12}$$

where $\frac{dW'}{dA}$ can be calculated as

$$-\eta_w \nabla^2_{A,W} L_{tgt}(W, A, D^{(tr)}) \tag{13}$$

Matrix-vector multiplication is approximated using finite-difference approximation similar to (Liu et al., 2019).

For $\nabla_A L_{tgt}(W', A, D^{(val)})$, it can be calculated as:

$$\nabla_A L_{tgt}(W', A, D^{(val)}) = \frac{dW'}{dA} \frac{\partial L_{tgt}(W', A, D^{(val)})}{\partial W'} + \frac{\partial L_{tgt}(W', A, D^{(val)})}{\partial A} \tag{14}$$

where $\frac{dW'}{dA}$ is given in Eq.(13).

$\nabla_B (L_{tgt}(W', A, f(G')))$ can be computed as

$$\nabla_B L_{tgt}(W', A, f(G')) \quad = \frac{dG'}{dB} \nabla_{G'} L_{tgt}(W', A, f(G')) \\ = -\eta_g \nabla^2_{B,G} L_{dg}(G, B, D^{(tr)}) \nabla_{G'} L_{tgt}(W', A, f(G')) \tag{15}$$

$\nabla_B L_{tgt}(V', D^{(val)})$ can be computed as

$$\nabla_B L_{tgt}(V', D^{(val)}) \quad = \frac{dG'}{dB} \frac{dV'}{dG'} \nabla_{V'} L_{tgt}(V', D^{(val)}) \\ = \eta_g \eta_v \nabla^2_{B,G} L_{dg}(G, B, D^{(tr)}) \nabla^2_{G',V} L_{tgt}(V, f(G')) \nabla_{V'} L_{tgt}(V', D^{(val)}) \tag{16}$$

The gradient descent updates of $A$ and $B$ in Eq.(10) and Eq.(11) can run one or more steps. After $A$ and $B$ are updated, the one-step gradient-descent approximations (in Eq.(7), Eq.(8), Eq.(9)), which are functions of $A$ and $B$, change with $A, B$ and need to be re-updated. Then, the gradients of $A$ and $B$, which are functions of one-step gradient-descent approximations, need to be re-calculated and are used to refresh $A, B$. In sum, the updates of $A, B$ and the updates of one-step gradient-descent approximations mutually depend on each other. These updates are performed iteratively until convergence. Algorithm 1 shows the algorithm. Similar to (Liu et al., 2019), for matrix-vector multiplication operations in Eq.(12), Eq.(15), and Eq.(16), we approximate them using finite differences to reduce computational costs.

---
**Algorithm 1** Optimization algorithm
---
**While** not converged
  1. Update the approximation $W'$ of $W^*(A)$ using Eq.(7)
  2. Update the approximation $G'$ of $G^*(B)$ using Eq.(8)
  3. Update the approximation $V'$ of $V^*(G^*(B))$ using Eq.(9)
  4. Update $A$ using Eq.(10)
  5. Update $B$ using Eq.(11)
---

## A.1 Discussion on optimization algorithm

The data generator $G$ is initialized using a pretrained FQ-GAN generator, which is therefore able to generate realistic images at the early stage in our algorithm. The auxiliary model $V$ is initialized using a pretrained ResNet-18 model, which has good classification performance on a validation set. If $V$'s validation performance is improved after $V$ is finetuned on generated examples, it implies that the generated examples are "meaningful" since they can help to train a better-performing classifier.

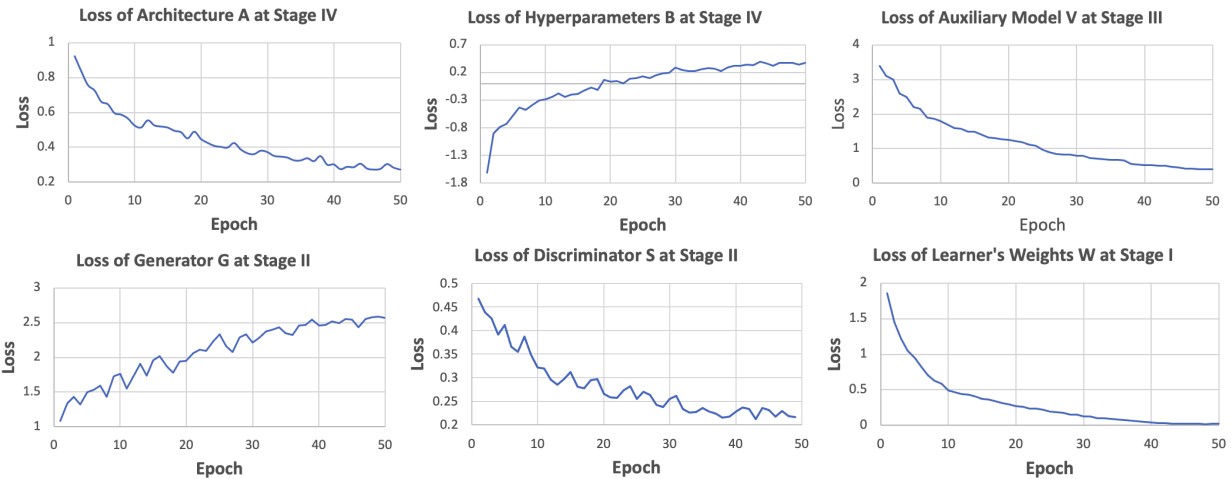

Figure 4: Loss curves of different variables in our method which is applied to Darts and searches for architectures on Cifar-10. Batch size is 64 and initial learning rate is 0.025.

The generator has weights parameters denoted by $G$ and hyperparameters denoted by $B$. For each training step, the updates of $W$ and weights $G$ can be conducted in parallel since they are not directly dependent on each other, as can be seen from Eq.(7) and Eq.(8). The generation of worst-case images is controlled by hyperparameters $B$. At each training step, to generate worst-case images for the current learner $W$, we first update $B$ by maximizing the validation loss of $W$ on generated data, using Eq.(11). Note that the update of $B$ depends on $W$. Afterwards, the generator with updated hyperparameters $B$ can generate worst-case images for $W$ since $B$ was updated to explicitly increase the validation loss of $W$ on generated data.

## A.2   Further discussion on computational efficiency

Optimizing the hyperparameters of the generative model is efficient. These hyperparameters are two continuous scalars, which can be efficiently optimized using gradient-based methods. Before optimizing them in our method, we initialized these hyperparameters using values reported in FQ-GAN, which further reduced convergence time.

## A.3   Stability of our method

In this section, we demonstrate the stability of methods from three perspectives.

**First, the objective function values in our optimization algorithm converge stably**, as shown in Figure 4, 5, and 6, where the loss curves are for optimization variables in our method which is applied to Darts and searches for architectures on CIFAR-10. The corresponding optimization problem is:

$$
\begin{aligned}
\min_A \max_B \quad & L_{tgt}(W^*(A), f(G^*(B), z_2)) - \lambda L_{tgt}(V^*(G^*(B)), D^{(val)}) \\
s.t. \quad & V^*(G^*(B)) = \text{argmin}_V \; L_{tgt}(V, f(G^*(B), z_1)) \\
& G^*(B) = \text{argmax}_G \min_S \; L_{gan}(G, B, S, D^{(tr)}) \\
& W^*(A) = \text{argmin}_W \; L_{tgt}(A, W, D^{(tr)})
\end{aligned}
\tag{17}
$$

At stage II, $L_{gan}(G, B, S, D^{(tr)})$ is a GAN loss, $S$ is the discriminator, $G$ is the generator, and $B$ are the hyperparameters of $G$. In this formulation, $G$ and $B$ are to be maximized and the rest of variables are to be minimized. From Figure 4, 5, and 6, we can see that the loss values of these variables converge stably.

**Second, the architectures searched by our method at increasing epochs achieve increasingly better accuracy on test data and the test accuracy of these architectures convergences stably**, as shown in Figure 7, where our method (Ours-Darts) was applied to Darts and the search was performed on CIFAR-10. This measure of stability is proposed in (Chen & Hsieh, 2020b; Bao et al., 2021; Xue et al., 2021).

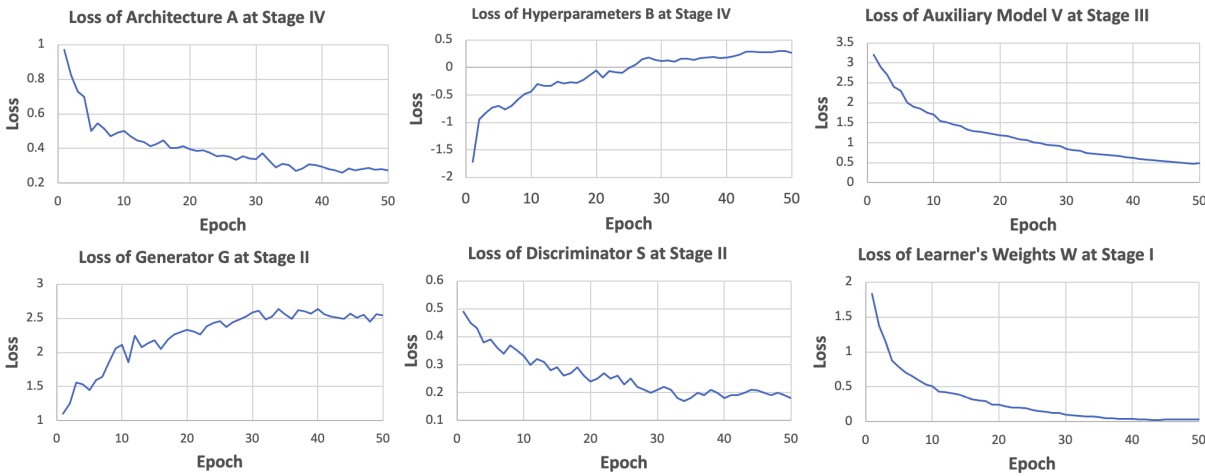

Figure 5: For batch size of 128 and initial learning rate of 0.025. Loss curves of different variables in our method which is applied to Darts and searches for architectures on Cifar-10.

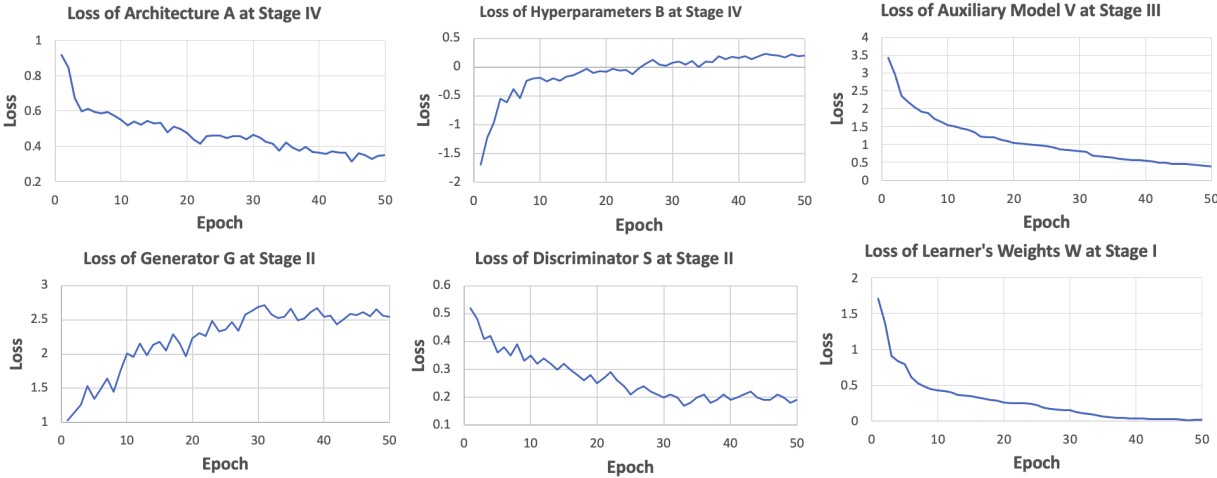

Figure 6: For batch size of 64 and initial learning rate of 0.05. loss curves of different variables in our method which is applied to Darts and searches for architectures on Cifar-10.

**Third, the optimization in our method converges stably under different hyperparameter settings and does not have high sensitivity to hyperparameters**, as shown in Figure 4, 5, and 6. In Figure 4, batch size is 64 and initial learning rate is 0.025. In Figure 5, batch size is 128 and initial learning rate is 0.025. In Figure 6, batch size is 64 and initial learning rate is 0.05. Our method was applied to Darts and the search was performed on CIFAR-10. As can be seen, under different hyperparameter values of batch size and initial learning rate, the losses in our optimization algorithm converge stably. In our experiments reported in the main paper, the values of hyperparameters (including batch size, learning rate, learning rate scheduler, etc.) mostly followed those used in baselines. Our algorithm converges stably under these default hyperparameter values without tuning.

# B Additional experimental results

## B.1 Results on ImageNet-A, ImageNet-R, and ImageNet-Sketch

We also evaluated our method on ImageNet-A (Hendrycks et al., 2021b), ImageNet-R (Hendrycks et al., 2021a), and ImageNet-Sketch (Wang et al., 2019). The experimental settings are the same as those described in Section 4.2 in the

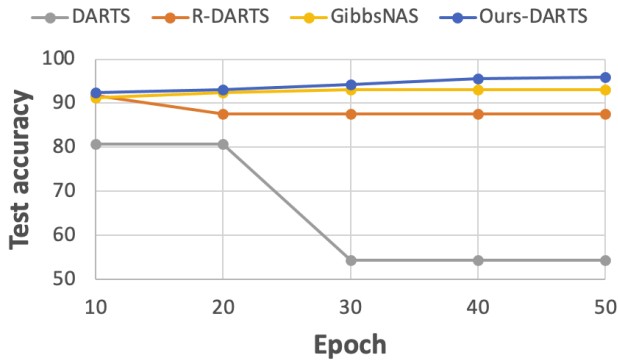

Figure 7: Test accuracy of architectures searched at different epochs. The experiments were performed on the search space of DARTS and on CIFAR-10.

main paper. Our framework was applied to PCDARTS. Table 10, 11, and 12 show the results. As can be seen, our method outperforms baselines, which further demonstrates the effectiveness of our method in searching for worst-case robust neural architectures.

Table 10: Accuracy on ImageNet-A. Architectures are searched on ImageNet.

| Method | Accuracy |
|---|---|
| Pcdarts | 6.2 |
| SPCL-Pcdarts | 6.6 |
| DIHCL-Pcdarts | 6.3 |
| Ours-Pcdarts | **10.1** |

Table 11: Top-1 error on ImageNet-R. Architectures are searched on ImageNet.

| Method | Accuracy |
|---|---|
| Pcdarts | 62.3 |
| SPCL-Pcdarts | 62.7 |
| DIHCL-Pcdarts | 61.9 |
| Ours-Pcdarts | **59.5** |

Table 12: Top-1 accuracy on ImageNet-Sketch. Architectures are searched on ImageNet.

| Method | Accuracy |
|---|---|
| Pcdarts | 12.8 |
| SPCL-Pcdarts | 12.9 |
| DIHCL-Pcdarts | 12.5 |
| Ours-Pcdarts | **16.1** |

## B.2 Additional ablation studies

### B.2.1 Perform ablation study by removing stages

One way of checking the effectiveness of each stage is to remove this stage from the framework and check how performance is affected. In our framework, there are four stages. The first stage and four stage are needed to perform architecture search. Therefore, they cannot be removed. Similar to other NAS methods, in the first stage, we train

weight parameters; in the fourth stage, we update architecture variables. Removing either of them will render the architectures unable to be searched.

To remove the second stage, we can use a pretrained GAN to generate data instead of training a GAN end-to-end in our framework. Please see the Pretain ablation study in Section B.2.2 for details.

To remove the third stage, we can simply set the tradeoff parameter $\lambda$ in our framework to zero, which yields the "adversarial only" ablation setting as studied in Section 4.6 in the main paper (please see that section for details).

### B.2.2 Ablation on the generative model of the tester

For the generative model of the tester, we perform the following ablation studies. The studies are performed on Ours-darts. The datasets are Cifar-100 and Cifar-10. The generative model is FQ-GAN (Zhao et al., 2020).

- **Weights-only**. We fix the meta parameters of FQ-GAN and only learn its weight parameters. The meta parameters of FQ-GAN are set to the default values in (Zhao et al., 2020). For the generator $G$ in FQ-GAN, we learn it at the second stage. For the discriminator $S$, we learn it at the fourth stage. Let $L_{fqgan}$ denote the FQ-GAN loss. The corresponding formulation is:

$$
\begin{aligned}
\min_A \max_S \quad & L_{tgt}(W^*(A), f(G^*(S), z_2)) - \lambda L_{tgt}(V^*(G^*(S)), D^{(val)}) \\
s.t. \quad & V^*(G^*(S)) = \operatorname{argmin}_V L_{tgt}(V, f(G^*(S), z_1)) \\
& G^*(S) = \operatorname{argmin}_G L_{fqgan}(G, S, D^{(tr)}) \\
& W^*(A) = \operatorname{argmin}_W L_{tgt}(A, W, D^{(tr)})
\end{aligned}
\tag{18}
$$

- **Pretrain**. We use a fixed-weights FQ-GAN pretrained by (Zhao et al., 2020) instead of learning it in our framework. We use the pretrained FQ-GAN to generate $M = 100K$ image-label pairs $\{(p_i, q_i)\}_{i=1}^{M}$ where $q_i$ is a label and $p_i$ is the corresponding image, then select an adversarial validation set from the 100K pairs. For each generated pair, we associate it with a selection variable $s \in [0, 1]$. The larger $s$ is, the more likely that the pair is selected. Let $\ell_{tgt}(W^*(A), p_i, q_i)$ denote the loss defined on the pair $(p_i, q_i)$. The corresponding formulation is:

$$
\begin{aligned}
\min_A \max_{\{s_i\}_{i=1}^M} \quad & \frac{1}{\sum_{i=1}^M s_i} \sum_{i=1}^M s_i \ell_{tgt}(W^*(A), p_i, q_i) + \lambda L_{tgt}(W^*(A), D^{(val)}) \\
s.t. \quad & W^*(A) = \operatorname{argmin}_W L_{tgt}(A, W, D^{(tr)})
\end{aligned}
\tag{19}
$$

At the second stage, we learn the selection variables $\{s_i\}_{i=1}^M$ by maximizing the learner's validation loss $\sum_{i=1}^M s_i \ell_{tgt}(W^*(A), p_i, q_i)$. We learn the architecture $A$ by minimizing the loss $\sum_{i=1}^M s_i \ell_{tgt}(W^*(A), p_i, q_i)$ on selected validation examples and minimizing the loss $L_{tgt}(W^*(A), D^{(val)})$ on the human-provided validation set $D^{(val)}$. $\lambda$ is a tradeoff parameter. $\frac{1}{\sum_{i=1}^M s_i}$ is a normalization term.

In our full method, both the meta parameters and weight parameters of FQ-GAN are learned. Table 13 shows the results. We make two observations. First, our full method works better than Weights-only. The reason is: in Weights-only, the values of meta parameters are fixed and these values may not be optimal for generating adversarial validation sets. In contrast, our full method learns these meta parameters by maximizing the validation loss and the meta parameters are optimized specifically for achieving the goal of generating high-fidelity adversarial validation sets. Second, our full method and Weights-only work better than Pretrain. The reason is: in Pretrain, weights parameters are fixed and they may not be optimal for generating adversarial validation sets. In our full method and Weights-only, the weights parameters are learned specifically for achieving the goal of generating high-fidelity adversarial validation sets.

**How meta parameters affect data generation.** In our method, the meta parameters of the generative model FQ-GAN include two hyperparameters: momentum decay $\lambda$ and FQ weight $\alpha$.

$\lambda$ controls how much history is leveraged for dictionary construction. The larger $\lambda$ is, the more history is leveraged. The influence of $\lambda$ on data generation performance is shown in Figure 4(c) in (Zhao et al., 2020). A value of $\lambda$ in the middle ground, which properly balances history and current statistics, yields the lowest FID score. The IS score remains roughly the same when $\lambda$ increases.

Table 13: Test errors on CIFAR-100 and CIFAR-10.

| Method | Cifar-100 | Cifar-10 |
|---|---|---|
| Weights-only | 19.42±0.21 | 2.71±0.02 |
| Pretrain | 19.77±0.15 | 2.74±0.03 |
| Our full method | **18.55**±0.09 | **2.68**±0.03 |

$\alpha$ is the tradeoff weight associated with the quantization loss in (Zhao et al., 2020). The larger $\alpha$ is, the more emphasis on quantization. The influence of $\alpha$ on data generation performance is shown in Figure 4(d) in (Zhao et al., 2020). As $\alpha$ increases, FID decreases and IS remains roughly the same.

### B.2.3 Training the generator from scratch

We trained the generator from scratch (within our multi-level optimization based method). Table 14 compares 1) Scratch: train the generator from scratch, and 2) Pretrain: first pretrain the generator, then finetune it within our method. The experiments were performed by applying these methods to Darts2nd and Pdarts. As can be seen, Scratch and Pretrain achieve similar performance, which demonstrates that the generator is not dominated by pretraining and finetuning plays a significant role.

Table 14: Experimental results for training the generator from scratch.

| Method | Error on Cifar100 | Error on Cifar10 |
|---|---|---|
| Scratch + Darts2nd | 18.59±0.11 | 2.70±0.04 |
| Pretrain + Darts2nd | 18.55±0.09 | 2.68±0.03 |
| Scratch +Pdarts | 16.18±0.06 | 2.48±0.05 |
| Pretrain + Pdarts | 16.17±0.08 | 2.48±0.04 |

### B.2.4 Sensitivity analysis of hyperparameters

Figure 8 and Figure 9 show how average classification error (%) on CIFAR-10-C (left) and mean corruption error (mCE, %) on ImageNet-C change with $\lambda$ and $\gamma$. For experiments on CIFAR-10-C, our framework was applied to DARTS2nd. For experiments on Image-C, our framework was applied to PCDARTS. As can be seen, values of $\lambda$ and $\gamma$ in the middle ground achieve the best performance. From the tester's perspective, $\lambda$ explores a tradeoff between maximizing the learner's validation loss and making generated validation examples meaningful. Increasing $\lambda$ encourages the tester to generate validation examples that are more meaningful. Validation examples with more meaningfulness can more reliably evaluate the learner. However, if $\lambda$ is too large, the validation examples are biased to be more meaningful but are less effective in increasing the learner's validation loss. Consequently, the generated validation examples may not be able to effectively measure the learner's worst-case performance and cannot drive the learner to improve its worst-case performance.

### B.3 Experiments on adversarial attack

We perform an evaluation of our method against adversarial attacks. Two datasets were utilized: CIFAR-10 and ImageNet. Our approach is compared with several baseline methods: RobNet (Guo et al., 2020), SDARTS-ADV (Chen & Hsieh, 2020b), AND PC-DARTS-ADV (Chen & Hsieh, 2020b). The robustness of our method is tested against three well-known untargeted white-box adversarial attacks: the fast gradient sign method (FGSM) (Goodfellow et al., 2014b), projected gradient descent (PGD) (Madry et al., 2017), and Carlini & Wagner (C&W) attack (Carlini & Wagner, 2017).

Our method was applied to PC-DARTS. The number of training epochs was set to 50. We used SGD to optimize the network weights, with a learning rate of 0.1, batch size of 256, momentum of 0.9, and weight decay of 3e-4. Architecture variables were optimized using the Adam optimizer with a static learning rate of 6e-4, $\beta_1$ of 0.5, $\beta_2$ of 0.999, and a weight decay of 3e-4. The final architecture was obtained by stacking 20 copies of the searched cell,

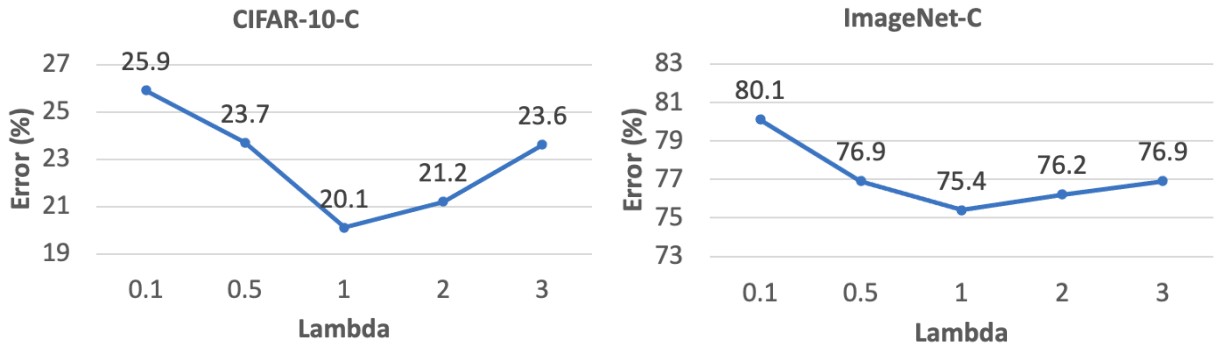

Figure 8: How average classification error (%) on CIFAR-10-C (left) and mean corruption error (mCE, %) on ImageNet-C change with $\lambda$.

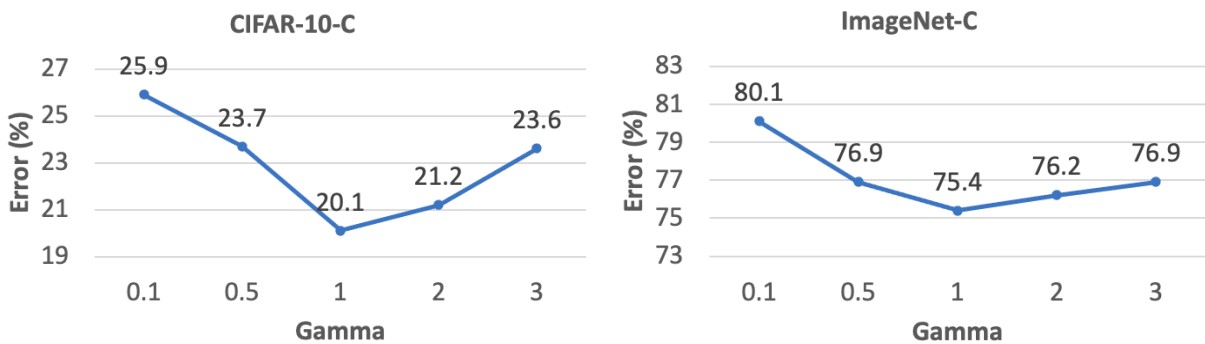

Figure 9: How average classification error (%) on CIFAR-10-C (left) and mean corruption error (mCE, %) on ImageNet-C change with $\gamma$.

which was then trained from scratch for 600 epochs, with a batch size of 128, an initial learning rate of 0.025, norm gradient clipping of 5, drop-path rate of 0.3, and 36 channels.

Table 15 and 16 show the results. As can be seen, our method is more robust to adversarial attacks, due to its mechanism of optimizing architecture variables by minimizing the loss on generated adversarial validation sets.

Table 15: Mean accuracy and standard deviation of five runs under various attacks on CIFAR-10. In PGD $(n)$, $n$ denotes the number of iterations.

| Method | PGD (10) | PGD (20) | PGD (100) | FGSM | C&W |
|---|---|---|---|---|---|
| RobNet-large (Guo et al., 2020) | 49.49 | 49.44 | 49.24 | 54.98 | 47.19 |
| RobNet-free (Guo et al., 2020) | 52.80 | 52.74 | 52.57 | 58.38 | 46.95 |
| SDARTS-ADV (Chen & Hsieh, 2020b) | 56.94 ± 0.02 | 56.90 ± 0.04 | 56.77 ± 0.17 | 63.84 ± 0.02 | 42.67 ± 0.09 |
| PC-DARTS-ADV (Chen & Hsieh, 2020b) | 57.15 ± 0.02 | 57.11 ± 0.05 | 56.83 ± 0.21 | 65.29 ± 0.03 | 42.58 ± 0.04 |
| Ours-pcdarts | **59.07 ± 0.04** | **59.14 ± 0.08** | **59.72 ± 0.11** | **68.30 ± 0.07** | **51.27 ± 0.09** |

## B.4 Evaluate the transferability of searched architectures to STL-10

In this section, we evaluate the transferability of searched architectures to the STL-10 (Coates et al., 2011) dataset. Darts2nd-Cifar10 denotes an architecture searched by Darts2nd on Cifar-10. Similar meanings hold for other notations in such a format. Given a searched architecture $A$, its transferability to STL-10 is investigated by performing architecture evaluation of $A$ on STL-10. Table 17 shows the accuracy on the test set of STL-10. Our methods achieve

Table 16: Mean accuracy and standard deviation of five runs under various attacks on ImageNet. In PGD $(n)$, $n$ denotes the number of iterations.

| Method | PGD (100) | FGSM | C&W |
|---|---|---|---|
| RobNet-large (Guo et al., 2020) | 37.14 | 39.74 | 25.73 |
| SDARTS-ADV (Chen & Hsieh, 2020b) | $46.54 \pm 0.13$ | $48.09 \pm 0.07$ | $36.86 \pm 0.10$ |
| PC-DARTS-ADV (Chen & Hsieh, 2020b) | $46.59 \pm 0.15$ | $48.25 \pm 0.08$ | $36.69 \pm 0.09$ |
| Ours-pcdarts | $\mathbf{47.82 \pm 0.11}$ | $\mathbf{50.66 \pm 0.10}$ | $\mathbf{40.37 \pm 0.14}$ |

better accuracy than baselines, which demonstrates that training on synthetic worst-case examples does not sacrifice transferability. These results are consistent with the transferability results on ImageNet which are reported in Table 23, where the architectures searched by our method on Cifar10/100 have better transferability to ImageNet than baselines. A possible reason is: our method searches for architectures that are robust against various challenging worst-case examples, which helps to preserve the transferability of these architectures to other datasets if we consider examples in other datasets as a special type of challenging examples (they are challenging because they are out of the data domain where architectures are searched).

Table 17: Test accuracy on STL-10.

| Architecture | Accuracy (%) |
|---|---|
| Darts2nd-Cifar10 | 86.5 |
| Ours-Darts2nd-Cifar10 | **88.9** |
| Darts2nd-Cifar100 | 86.1 |
| Ours-Darts2nd-Cifar100 | **88.6** |
| Pcdarts-Cifar10 | 87.2 |
| Ours-Pcdarts-Cifar10 | **90.4** |

## B.5 Additional experimental results for Table 1 and 5 in the main paper

Table 18 and 19 shows additional experimental results for Table 1 and 5 in the main paper.

Table 18: Results on CIFAR-100, including classification error (%) on the entire test set, number of parameters (millions) in the searched architecture, and search cost (GPU days). Ours-DARTS$^+$ denotes that our method is applied to the search space of DARTS$^+$. Similar meanings hold for other notations in such a format. DARTS-1st and DARTS-2nd denotes that first order and second order approximation is used in DARTS. * means the results are taken from DARTS$^-$ (Chu et al., 2020a). † means we re-ran this method for 10 times. $\Delta$ means the algorithm ran for 600 epochs instead of 2000 epochs in the architecture evaluation stage, to ensure a fair comparison with other methods (where the epoch number is 600). The search cost is measured by GPU days on a Tesla v100.

| Method | Error(%) | Param(M) | Cost |
|---|---|---|---|
| *DARTS$^-$ (Chu et al., 2020a) | $17.51 \pm 0.25$ | 3.3 | 0.4 |
| †DARTS$^-$ (Chu et al., 2020a) | $18.97 \pm 0.16$ | 3.1 | 0.4 |
| Ours-DARTS$^-$ | $18.29 \pm 0.10$ | 3.3 | 0.7 |
| $\Delta$DARTS$^+$ (Liang et al., 2019) | $17.11 \pm 0.43$ | 3.8 | 0.2 |
| Ours-DARTS$^+$ | $\mathbf{16.57 \pm 0.11}$ | 3.7 | 0.2 |

## B.6 Evaluation of generated images

We perform an automatic evaluation of images generated by our method Ours-darts2nd trained on CIFAR-100, using metrics including inception score (Salimans et al., 2016) and Frechet inception distance (FID) (Heusel et al., 2017). Table 20 shows the results. Our method outperforms vanilla FQ-GAN and BigGAN. The reason is: the images

Table 19: Results on CIFAR-10. * means the results are taken from DARTS$^-$ (Chu et al., 2020a), NoisyDARTS (Chu et al., 2020b), and DrNAS (Chen et al., 2020). The rest notations are the same as those in Table 5 in the main paper.

| Method | Error(%) | Param(M) | Cost |
|---|---|---|---|
| *DARTS-1st (Liu et al., 2019) | 3.00±0.14 | 3.3 | 0.4 |
| Ours-DARTS1st | **2.83**±0.08 | 2.6 | 0.7 |
| *DARTS$^-$ (Chu et al., 2020a) | 2.59±0.08 | 3.5 | 0.4 |
| $^\dagger$DARTS$^-$ (Chu et al., 2020a) | 2.97±0.04 | 3.3 | 0.4 |
| Ours-DARTS$^-$ | 2.77±0.05 | 3.2 | 0.4 |
| $^\triangle$DARTS$^+$ (Liang et al., 2019) | 2.83±0.05 | 3.7 | 0.4 |
| Ours-DARTS$^+$ | **2.68**±0.08 | 3.7 | 0.4 |

generated by our method are explicitly encouraged to be "meaningful". They are used to train an auxiliary model, which is then evaluated on a human-labeled validation set. If generated images have poor quality, the auxiliary model trained by them will perform poorly on the human-labeled validation set. Our framework prevents this from happening by explicitly minimizing the loss on the human-labeled validation set. Such a mechanism is lacking in vanilla FQ-GAN and BigGAN. Figure 2 in the main paper shows some randomly sampled images generated by our method. As can be seen, these images are realistic.

Table 20: Automatic evaluation of generated CIFAR-100 images.

| Method | Inception↑ | FID↓ |
|---|---|---|
| BigGAN | 9.36±0.10 | 9.01±0.44 |
| FQ-GAN | 9.59±0.04 | 7.42±0.07 |
| Ours | **9.75**±0.03 | **6.01**±0.09 |

## B.7 Significance test results

To check whether the performance of our method is significantly better than that of baselines, we perform statistical significance tests between the result of our method and the result of the corresponding baseline, using double-sided T-test. We use the function in the python package "scipy.stats.ttest_1samp" and report the average results over 10 different runs. Table 21 and 22 show the results.

Table 21: Significance test results on CIFAR-100

| Our method | Baseline | p-value |
|---|---|---|
| Ours-darts2nd | DIHCL-darts2nd | 5.61e-3 |
| Ours-darts2nd | SPCL-darts2nd | 6.63e-5 |
| Ours-darts2nd | Darts2nd | 6.15e-7 |
| Ours-pcdarts | DIHCL-pcdarts | 2.42e-5 |
| Ours-pcdarts | SPCL-pcdarts | 9.09e-5 |
| Ours-pcdarts | Pcdarts | 5.33e-5 |
| Ours-pdarts | DIHCL-pdarts | 8.85e-7 |
| Ours-pdarts | SPCL-pdarts | 2.36e-6 |
| Ours-pdarts | Pdarts | 1.46e-6 |
| Ours-prdarts | DIHCL-prdarts | 1.20e-4 |
| Ours-prdarts | SPCL-prdarts | 2.31e-3 |
| Ours-prdarts | Prdarts | 6.58e-3 |

From these two tables, we can see that the p-values are small between baselines methods and our methods, which demonstrate that the errors of our methods are significantly lower than those of baselines.

Table 22: Significance test results on CIFAR-10

| Our method | Baseline | p-value |
|---|---|---|
| Ours-darts2nd | DIHCL-darts2nd | 5.37e-8 |
| Ours-darts2nd | SPCL-darts2nd | 6.61e-7 |
| Ours-darts2nd | Darts2nd | 1.85e-6 |
| Ours-pcdarts | DIHCL-pcdarts | 8.05e-6 |
| Ours-pcdarts | SPCL-pcdarts | 5.59e-4 |
| Ours-pcdarts | Pcdarts | 9.72e-4 |
| Ours-pdarts | DIHCL-pdarts | 7.04e-9 |
| Ours-pdarts | SPCL-pdarts | 3.55e-7 |
| Ours-pdarts | Pdarts | 1.69e-6 |
| Ours-prdarts | DIHCL-prdarts | 6.29e-5 |
| Ours-prdarts | SPCL-prdarts | 2.50e-4 |
| Ours-prdarts | Prdarts | 7.81e-3 |

## B.8 Model parameters, search costs, and FLOPs on ImageNet

Table 23 shows the number of model parameters, search costs, and FLOPs on ImageNet. The parameter numbers, search costs, and FLOPs of our methods are close to those in differentiable baselines.

# C Additional discussions

## C.1 Difference between the discriminator and the auxiliary model

The discriminator $D$ in the generative model distinguishes whether an example is real or fake. The auxiliary model $V$ is utilized to evaluate whether generated data can train a well-performing classifier. $D$ is trained on generated data and real data by solving a binary classification problem (classifying whether an example is generated or real). $V$ is trained on generated data by solving a multi-class classification problem (e.g., classifying a generated example into one of the 10 classes in CIFAR-10 or one of the 100 classes in CIFAR-100).

## C.2 Discussion on bi-level optimization and GAN

GAN has been formulated as a bi-level optimization (BLO) problem in (Pfau & Vinyals, 2016). In the lower-level optimization problem, the optimization variable is the generator and the objective function is the probability that a generated example is classified by the discriminator as being a real example. The objective function is to be maximized w.r.t the generator. The discriminator is tentatively fixed at the lower-level. In the upper-level optimization problem, the optimization variable is the discriminator and the objective function is the regular GAN loss, which is a cross-entropy loss for distinguishing real examples from generated examples. The optimally trained generator at the lower-level, which is a function of the discriminator, is plugged into the cross-entropy loss at the upper-level. Please refer to Section 2.1 in (Pfau & Vinyals, 2016) for mathematical details.

To apply our method to GAN, we can set the learner to be a GAN model, where $W$ is the generator and $A$ is the discriminator. The optimization problem at stage I is the same as the lower-level optimization problem in the BLO formulation of GAN. The first loss term at stage IV is the same as the upper-level loss in the BLO formulation of GAN.

# D Additional experimental details

## D.1 Hyperparameter tuning

Setting hyperparameters for our algorithm is easy and light-weight. Our algorithm does not need extensive hyperparameter tuning. In our training algorithm, for hyperparameters such as learning rate, we simply followed their default

Table 23: Results on ImageNet, including top-1 and top-5 classification errors on the test set, number of weight parameters (millions), search cost (GPU days), and FLOPs (M). * means the results are taken from DARTS⁻ (Chu et al., 2020a) and DrNAS (Chen et al., 2020). The rest notations are the same as those in Table 5 in the main paper. The first row block shows networks designed by humans manually. The second row block shows non-gradient based search methods. The third block shows gradient-based methods. ‡ means the results following the hyperparameters selected for CIFAR10/100. The hyperparameter for CIFAR100 is used when directly searching on ImageNet.

| Method | Top-1 Error (%) | Top-5 Error (%) | Param (M) | Cost (GPU days) | FLOPs (M) |
|---|---|---|---|---|---|
| *Inception-v1 (Szegedy et al., 2015) | 30.2 | 10.1 | 6.6 | - | 1448 |
| *MobileNet (Howard et al., 2017) | 29.4 | 10.5 | 4.2 | - | 569 |
| *ShuffleNet 2× (v1) (Zhang et al., 2018) | 26.4 | 10.2 | 5.4 | - | 524 |
| *ShuffleNet 2× (v2) (Ma et al., 2018) | 25.1 | 7.6 | 7.4 | - | 299 |
| *NASNet-A (Zoph et al., 2018) | 26.0 | 8.4 | 5.3 | 1800 | 564 |
| *PNAS (Liu et al., 2018a) | 25.8 | 8.1 | 5.1 | 225 | 588 |
| *MnasNet-92 (Tan et al., 2019) | 25.2 | 8.0 | 4.4 | 1667 | 388 |
| *AmoebaNet-C (Real et al., 2019) | 24.3 | 7.6 | 6.4 | 3150 | 570 |
| *SNAS-CIFAR10 (Xie et al., 2019) | 27.3 | 9.2 | 4.3 | 1.5 | 522 |
| *BayesNAS-CIFAR10 (Zhou et al., 2019) | 26.5 | 8.9 | 3.9 | 0.2 | - |
| *PARSEC-CIFAR10 (Casale et al., 2019) | 26.0 | 8.4 | 5.6 | 1.0 | - |
| *GDAS-CIFAR10 (Dong & Yang, 2019) | 26.0 | 8.5 | 5.3 | 0.2 | 581 |
| *DSNAS-ImageNet (Hu et al., 2020) | 25.7 | 8.1 | - | - | 324 |
| *SDARTS-ADV-CIFAR10 (Chen & Hsieh, 2020a) | 25.2 | 7.8 | 5.4 | 1.3 | - |
| *PC-DARTS-CIFAR10 (Xu et al., 2020) | 25.1 | 7.8 | 5.3 | 0.1 | 586 |
| *ProxylessNAS-ImageNet (Cai et al., 2019) | 24.9 | 7.5 | 7.1 | 8.3 | 465 |
| *FairDARTS-CIFAR10 (Chu et al., 2019) | 24.9 | 7.5 | 4.8 | 0.4 | 386 |
| *FairDARTS-ImageNet (Chu et al., 2019) | 24.4 | 7.4 | 4.3 | 3.0 | 440 |
| *DrNAS-ImageNet (Chen et al., 2020) | 24.2 | 7.3 | 5.2 | 3.9 | - |
| *DARTS⁺-ImageNet (Liang et al., 2019) | 23.9 | 7.4 | 5.1 | 6.8 | 582 |
| *DARTS⁻-ImageNet (Chu et al., 2020a) | 23.8 | 7.0 | 4.9 | 4.5 | 467 |
| *DARTS⁺-CIFAR100 (Liang et al., 2019) | 23.7 | 7.2 | 5.1 | 0.2 | 591 |
| *DARTS2nd-CIFAR10 (Liu et al., 2019) | 26.7 | 8.7 | 4.7 | 4.0 | 574 |
| Ours-DARTS2nd-CIFAR10 | 25.8 | 8.2 | 4.7 | 3.9 | 554 |
| *P-DARTS (CIFAR10) (Chen et al., 2019) | 24.4 | 7.4 | 4.9 | 0.3 | 557 |
| ‡Ours-PDARTS-CIFAR10 | 24.1 | 7.1 | 4.9 | 0.3 | 541 |
| *P-DARTS (CIFAR100) (Chen et al., 2019) | 24.7 | 7.5 | 5.1 | 0.3 | 577 |
| ‡Ours-PDARTS-CIFAR100 | 24.0 | 7.1 | 5.1 | 0.3 | 595 |
| *PC-DARTS-ImageNet (Xu et al., 2020) | 24.2 | 7.3 | 5.3 | 3.8 | 597 |
| ‡Ours-PCDARTS-ImageNet | **23.5** | **6.7** | 5.2 | 3.7 | 569 |

values given in the DARTS baseline without tuning them. For example, the learning rate of weight parameters (including $W$, $G$, $V$) was simply set to the learning rate of weight parameters in DARTS; the learning rate of meta parameters (including $A$ and $B$) was simply set to the learning rate of architecture variables in DARTS. As shown in Figure 4, our algorithm converged very well under these default hyperparameter values. Besides, our algorithm is not sensitive to these hyperparameters and converges stably under different hyperparameter values, as shown in Figures 4, 5, and 6.

To tune the hyperparameter $\lambda$, we randomly sample 2.5K data from the 25K training set and sample 2.5K data from the 25K validation set. Then we use the 5K sampled data as a hyperparameter tuning set. $\lambda$ is tuned in $\{0.1, 0.5, 1, 2, 3\}$. For each configuration of $\lambda$, we use the remaining 22.5K training data and 22.5K validation data to perform architecture search and use their combination to perform architecture evaluation (retraining a larger stacked network from scratch). Then we measure the performance of the stacked network on the 5K sampled data. $\lambda$ value yielding the best performance on the 5K sampled data is selected. For other hyperparameters, they mostly follow those in DARTS (Liu

et al., 2019), P-DARTS (Chen et al., 2019), PC-DARTS (Xu et al., 2020), and PR-DARTS (Zhou et al., 2020a). The final selected hyperparameters are in Section E.

To obtain the best architecture during architecture evaluation, we split the 50k training set of CIFAR-10/100 to a 5k validation set for performance evaluation and a 45k set for training. The test set is never used for architecture selection or training.

## D.2 Experimental details of neural architecture search

### D.2.1 DARTS2nd based experiments

For methods based on DARTS2nd, including Ours-darts2nd, DIHCL-darts2nd, SPCL-darts2nd, the experimental settings are similar. In search spaces of DARTS, the candidate operations include: $3 \times 3$ and $5 \times 5$ separable convolutions, $3 \times 3$ and $5 \times 5$ dilated separable convolutions, $3 \times 3$ max pooling, $3 \times 3$ average pooling, identity, and zero. The stride of all operations is set to 1. The convolved feature maps are padded to preserve their spatial resolution. The order for convolutional operations is ReLU-Conv-BN. Each separable convolution is applied twice. The convolutional cell has 7 nodes. The output node is the depthwise concatenation of all intermediate nodes, excluding the input nodes. The first and second nodes of cell $k$ are equal to the outputs of cell $k-2$ and cell $k-1$, respectively. 1×1 convolutions are inserted when necessary. Reduction cells are located at the 1/3 and 2/3 of the total depth of the network. In reduction cells, operations adjacent to the input nodes have a stride of 2.

For CIFAR-10 and CIFAR-100, during architecture search, the learner's network is a stack of 8 cells, with the initial channel number set to 16. Each cell contains 7 nodes. The search is performed for 50 epochs, with a batch size of 64. The learner's network weights $W$ were optimized using SGD with a learning rate of 0.025, a momentum of 0.9, and a weight decay of 0.0003. The architecture variables $A$ were optimized using Adam (Kingma & Ba, 2014) with a learning rate of 0.001, a momentum of $(0.5, 0.999)$, and a weight decay of 0.001. The learning rate was scheduled with cosine scheduling. The architecture variables were initialized with zero initialization.

During architecture evaluation on CIFAR-10 and CIFRA-100, 20 copies of the searched cell are stacked to form the learner's network, with the initial channel number set to 36. The network is trained for 600 epochs with a batch size of 96 (for both CIFAR-10 and CIFAR-100). The SGD optimizer is used with an initial learning rate of 0.025 (annealed down to zero following a cosine schedule without restart), a momentum of 0.9, a weight decay of $3e - 4$, and a norm gradient clipping at 5. Drop-path with a rate of 0.3 as well as cutout is also used for regularization. Cutout, path dropout of probability 0.2 and auxiliary towers with weight 0.4 were applied.

For architecture evaluation on ImageNet, following (Liu et al., 2019), we take the architecture searched on CIFAR-10 and evaluate it on ImageNet. 14 cells (searched on CIFAR-10) are stacked to form a large network and the initial channel number was set as 48. The network is trained for 250 epochs with a batch size of 1024. Each experiment is repeated for ten times with the random seed to be from 1 to 10. We report the mean and standard deviation of results obtained from the 10 runs. For ImageNet experiments, we apply the mobile setting where the input image size is fixed to be $224 \times 224$ and the number of multi-add operations does not exceed 600M in the testing stage.

### D.2.2 PC-DARTS based experiments

For methods based on PC-DARTS, including Ours-pcdarts, DIHCL-pcdarts, SPCL-pcdarts, the experimental settings are similar. The search space of PC-DARTS follows that of DARTS. For architecture search on CIFAR-100 and CIFAR-10, the hyperparameter $K$ was set to 4. The network is a stack of 8 cells. Each cell contains 6 nodes. Initial channel number is set to 16. The architecture variables are trained using the Adam optimizer for 50 epochs. The learning rate is set to $6e - 4$, without decay. The weight decay is set to $1e - 3$. The momentum is set to $(0.5, 0.999)$. The network weight parameters are trained using SGD for 50 epochs. The initial learning rate is set to 0.1. Cosine scheduling is used to decay the learning rate, down to 0 without restart. The momentum is set to 0.9. The weight decay is set to $3e - 4$. The batch size is set to 256. Warm-up is utilized: in the first 15 epochs, architecture variables are frozen and only network weights are optimized.

The settings for architecture evaluation on CIFAR-100 and CIFAR-10 follow those of DARTS. 18 normal cells and 2 reduction cells are stacked into a large network. The initial channel number is set to 36. The stacked network is trained from scratch using SGD for 600 epochs, with batch size 128, initial learning rate 0.025, momentum 0.9, weight

decay $3e - 4$, norm gradient clipping 5, drop-path rate 0.3, and cutout. The learning rate is decayed to 0 using cosine scheduling without restart.

We combine our method and PC-DARTS to directly search for architectures on ImageNet. The stacked network starts with three convolution layers which reduce the input image resolution from 224×224 to 28×28, using stride 2. After the three convolution layers, 6 normal cells and 2 reduction cells are stacked. Each cell consists of $N = 6$ nodes. The sub-sampling rate was set to 0.5. The network was trained for 50 epochs. Architecture variables are trained using Adam. The learning rate is fixed to $6e - 3$. The weight decay is set to $1e - 3$. The momentum is set to $(0.5, 0.999)$. In the first 35 epochs, architecture variables are frozen. Network weight parameters are trained using SGD. The initial learning rate is set to 0.5. It is decayed to 0 using cosine scheduling without restart. Momentum is set to 0.9. Weight decay is set to $3e - 5$. The batch-size is set to 1024. Epoch number is set to 250. Eight Tesla V100 GPUs were used.

For architecture evaluation on ImageNet, the stacked network starts with three convolution layers which reduce the input image resolution from 224×224 to 28×28, using stride 2. After the three convolution layers, 12 normal cells and 2 reduction cells are stacked. Initial channel number is set to 48. The network is trained from scratch using SGD for 250 epochs, with batch size 1024, initial learning rate 0.5, weight decay $3e - 5$, and momentum 0.9. For the first 5 epochs, learning rate warm-up is used. The learning rate is linearly decayed to 0. Label smoothing and auxiliary loss tower is used.

### D.2.3 P-DARTS based experiments

The search process has three stages. At the first stage, the search space and stacked network in P-DARTS are mostly the same as DARTS. The only difference is the number of cells in the stacked network in P-DARTS is set to 5. At the second stage, the number of cells in the stacked network is 11. At the third stage, the cell number is 17. At stage 1, 2, 3, the initial Dropout probability on skip-connect is 0, 0.4, and 0.7 for CIFAR-10, is 0.1, 0.2, and 0.3 for CIFAR-100; the size of operation space is 8, 5, 3, respectively. The final searched cell is limited to have 2 skip-connect operations at maximum. At each stage, the network is trained using the Adam optimizer for 25 epochs. The batch size is set to 96. The learning rate is set to 6e-4. Weight decay is set to 1e-3. Momentum is set to $(0.5, 0.999)$. In the first 10 epochs, architecture variables are frozen and only network weights are optimized.

For architecture evaluation on CIFAR-100 and CIFAR-10, the stacked network consists of 20 cells. The initial channel number is set to 36. The network is trained from scratch using SGD. The epoch number is set to 600. The batch size is set to 128. The initial learning rate is set to 0.025. The learning rate is decayed to 0 using cosine scheduling. Weight decay is set to 3e-4 for CIFAR-10 and 5e-4 for CIFAR-100. Momentum is set to 0.9. Drop-path probability is set to 0.3. Cutout regularization length is set to 16. Auxiliary towers of weight 0.4 are used.

For architecture evaluation on ImageNet, the settings are similar to those of DARTS. The network consists of 14 cells. The initial channel number is set to 48. The network is trained from scratch using SGD for 250 epochs. Batch size is set to 1024. Initial learning rate is set to 0.5. The learning rate is linearly decayed after each epoch. In the first 5 epochs, learning rate warmup is used. The momentum is set to 0.9. The weight decay is set to $3e - 5$. Label smoothing and auxiliary loss tower are used during training. The network was trained on 8 Nvidia Tesla V100 GPUs.

### D.2.4 PR-DARTS based experiments

The operations include: 3×3 and 5×5 separable convolutions, 3×3 and 5×5 dilated separable convolutions, 3×3 average pooling and 3×3 max pooling, zero, and skip connection. The stacked network consists of $k$ cells. The $k/3$- and $2k/3$-th cells are reduction cells. In reduction cells, all operations have a stride of two. The rest cells are normal cells. Operations in normal cells have a stride of one. Cells of the same type (either reduction or normal) have the same architecture. The inputs of each cell are the outputs of two previous cells. Each cell contains four intermediate nodes and one output node. The output node is a concatenation of all intermediate nodes.

For architecture search on CIFAR-100 and CIFAR-10, the stacked network consists of 8 cells. The initial channel number is set to 16. In PR-DARTS, $\lambda_1$, $\lambda_2$, and $\lambda_3$ are set to 0.01, 0.005, and 0.005 respectively. The network was trained for 200 epochs. The mini-batch size is set to 128. Architecture variables are trained using Adam. The learning rate is set to $3e - 4$. The weight decay is set to $1e - 3$. Network weights are trained using SGD. The initial learning rate is set to 0.025. The momentum is set to 0.9. The weight decay is set to $3e - 4$. The learning rate is decayed to 0 using cosine scheduling. For acceleration, per iteration, only two operations on each edge are randomly selected to

update. The temperature $\tau$ is set to 10 and is linearly reduced to 0.1; $a = -0.1$ and $b = 1.1$. Pruning on each node is conducted by comparing the gate activation probabilities of all non-zero operations collected from all previous nodes and retaining top two operations.

For architecture evaluation on CIFAR10 and CIFAR100, the stacked network consists of 18 normal cells and 2 reduction cells. The initial channel number is set to 36. The network is trained from scratch using SGD. The mini-batch size is set to 128. The epoch number is set to 600. The initial learning rate is set to 0.025. The momentum is set to 0.9. The weight decay is set to $3e - 4$. The gradient norm clipping is set to 5. The drop-path probability is set to 0.2. The cutout length is set to 16. The learning rate is decayed to 0 using cosine scheduling.

For architecture evaluation on ImageNet, the input images are resized to $224 \times 224$. The stacked network consists of 3 convolutional layers, 12 normal cells, and 2 reduction cells. The channel number is set to 48. The network is trained using SGD for 250 epochs. The batch size is set to 128. The learning rate is set to 0.025. The momentum is set to 0.9. The weight decay is set to $3e - 4$. The gradient norm clipping is set to 5. The learning rate is decayed to 0 via cosine scheduling.

### D.2.5 Implementation details

We use PyTorch to implement all models. The version of Torch is 1.4.0 (or above). We build our method upon official python packages for different differentiable search approaches, such as "DARTS[1]", "P-DARTS[2]" and "PC-DARTS[3]".

### D.3 Experimental details of Figure 1(left) in the main paper

On the 25K training dataset of CIFAR-100, we train an ResNet-18 model. Hyperparameters are the same as those in (He et al., 2016a). Given the validation examples generated at epoch 10, 20, 30, 40, 50, we apply the trained ResNet-18 to these examples and measure the averaged errors (normalized by the number of examples), which are shown in Figure 1(left) in the main paper.

### D.4 Experimental details of Table 1 in the main paper

We asked three undergraduate students to select 2000 corner-case examples from the 10K test set of CIFAR-100 (20 examples per class). Given a randomly sampled image, the undergraduates were asked to label whether the class label of this image is difficult to recognize. If at least two undergraduates think this image is difficult to recognize, the image is labeled as a corner-case one. This procedure was repeated until 20 corner-case examples were selected for each class. Then we measure prediction errors of different methods on the 2000 corner-case examples. The results are given in Table 1 in the main paper.

### D.5 Experimental details on ablation studies

The formulation of "adversarial only" is:

$$\min_A \max_B \quad L_{tgt}(W^*(A), f(G^*(B), z_2))$$

$$s.t. \quad G^*(B) = \operatorname{argmin}_G L_{dg}(G, B, D^{(tr)}) \tag{20}$$

$$W^*(A) = \operatorname{argmin}_W L_{tgt}(A, W, D^{(tr)})$$

### D.6 Experimental details of evaluating robustness against overfitting

The four search spaces $S1 - S4$ are designed by (Zela et al., 2020).

- **S1**: In this search space, each edge has only two candidate operations. To identify these operations, operations that have the least importance in the original search space of DARTS are iteratively removed.

---

[1] https://github.com/quark0/darts
[2] https://github.com/chenxin061/pdarts
[3] https://github.com/yuhuixu1993/PC-DARTS/

- **S2**: For each edge, the candidate operations are 3×3 SepConv and SkipConnect.

- **S3**: For each edge, the candidate operations are: 3×3 SepConv, SkipConnect, and Zero.

- **S4**: For each edge, the candidate operations are: 3×3 SepConv and Noise. In the Noise operation, every value from the input feature map is replaced with random variables sampled from univariate Gaussian distribution.

## D.7 Instructions given to participants in human studies

Figure 10 shows the screenshot of instructions given to participants in human studies. We asked three undergraduates to label whether an image is difficult to recognize. Majority vote is taken to determine the final label (regarding whether an image is difficult to recognize).

For each image, please try to assign a class label to it. Is it easy to recognize the content of this image and assign the corresponding label to it? Please select Yes or No.

Here are two examples.

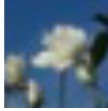 Yes

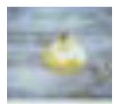 No

Figure 10: Screenshot of instructions given to participants in human studies.

## E   Full lists of hyperparameter settings

Table 25 to Table 32 show the hyperparameter settings used in different experiments in the search phase. Table 33 to Table 36 show the hyperparameter settings used in different experiments in the evaluation phase. Notations used in these tables are given in Table 24.

Table 24: Notations in our method

| Notation | Meaning |
|---|---|
| $A$ | Meta parameter of the learner |
| $W$ | Network weights of the learner |
| $B$ | Meta parameter of the data generation model |
| $G$ | Network weights of the data generation model |
| $V$ | Network weights of the auxiliary model |
| $D^{(\text{tr})}$ | Training data |
| $D^{(\text{val})}$ | Validation data |

Table 25: Hyperparameter settings in Ours-DARTS and Ours-DARTS$^+$ on CIFAR-10/100 during architecture search

| Name | Value |
|---|---|
| Optimizer for W,G,V | SGD |
| Initial learning rate for W,G,V | 0.025 |
| Learning rate scheduler for W,G,V | Cosine decay |
| Minimum learning rate for W,G,V | 0.001 |
| Momentum for W,G,V | 0.9 |
| Weight decay for W,G,V | 0.0003 |
| Optimizer for A,B | Adam |
| Learning rate for A,B | 0.0003 |
| Weight decay for A,B | 0.001 |
| Initial channels for W | 16 |
| Layers for W | 8 |
| Gradient Clip for W | 5 |
| Batch size | 64 |
| Epochs | 50 |
| $\lambda$ | 1 |

Table 26: Hyperparameter settings in Ours-DARTS$^-$ on CIFAR-10/100 during architecture search

| Name | Value |
|---|---|
| Optimizer for W,G,V | SGD |
| Initial learning rate for W,G,V | 0.025 |
| Learning rate scheduler for W,G,V | Cosine decay |
| Minimum learning rate for W,G,V | 0.001 |
| Momentum for W,G,V | 0.9 |
| Weight decay for W,G,V | 0.0003 |
| Optimizer for A,B | Adam |
| Learning rate for A,B | 0.0003 |
| Weight decay for A,B | 0.001 |
| Initial channels for W | 16 |
| Layers for W | 8 |
| Gradient Clip for W | 5 |
| Batch size | 64 |
| Epochs | 50 |
| $\lambda$ | 1 |
| $\beta$ for auxiliary skip connection | 1 |

Table 27: Hyperparameter settings in Ours-P-DARTS on CIFAR-10 during architecture search

| Name | Value |
|---|---|
| Optimizer for W,G,V | SGD |
| Initial learning rate for W,G,V | 0.025 |
| Learning rate scheduler for W,G,V | Cosine decay |
| Minimum learning rate for W,G,V | 0.0 |
| Momentum for W,G,V | 0.9 |
| Weight decay for W,G,V | 0.0003 |
| Optimizer for A,B | Adam |
| Learning rate for A | 0.0006 |
| Learning rate for B | 0.0003 |
| Weight decay for A,B | 0.001 |
| Initial channels for W | 16 |
| Layers for W | 8 |
| Gradient Clip for W,G,V,A | 5 |
| Batch size | 96 |
| Epochs | 25 |
| Add layers | [6,12] |
| Dropout rate | [0.1,0.4,0.7] |
| $\lambda$ | 0.5 |

Table 28: Hyperparameter settings in Ours-PDARTS on CIFAR-100 during architecture search

| Name | Value |
|---|---|
| Optimizer for W,G,V | SGD |
| Initial learning rate for W,G,V | 0.025 |
| Learning rate scheduler for W,G,V | Cosine decay |
| Minimum learning rate for W,G,V | 0.0 |
| Momentum for W,G,V | 0.9 |
| Weight decay for W,G,V | 0.0003 |
| Optimizer for A,B | Adam |
| Learning rate for A | 0.0006 |
| Learning rate for B | 0.0003 |
| Weight decay for A,B | 0.001 |
| Initial channels for W | 16 |
| Layers for W | 8 |
| Gradient Clip for W,G,V,A | 5 |
| Batch size | 96 |
| Epochs | 25 |
| Add layers | [6,12] |
| Dropout rate | [0.1,0.4,0.7] |
| $\lambda$ | 1 |

Table 29: Hyperparameter settings in Ours-PCDARTS on CIFAR-10 during architecture search

| Name | Value |
|---|---|
| Optimizer for W,G,V | SGD |
| Initial learning rate for W,G,V | 0.1 |
| Learning rate scheduler for W,G,V | Cosine decay |
| Minimum learning rate for W,G,V | 0.0 |
| Momentum for W,G,V | 0.9 |
| Weight decay for W,G,V | 0.0003 |
| Optimizer for A,B | Adam |
| Learning rate for A | 0.0006 |
| Learning rate for B | 0.0003 |
| Weight decay for A,B | 0.001 |
| Initial channels for W | 16 |
| Layers for W | 8 |
| Gradient Clip for W | 5 |
| Batch size | 256 |
| Epochs | 50 |
| $\lambda$ | 3 |

Table 30: Hyperparameter settings in Ours-PCDARTS on CIFAR-100 during architecture search

| Name | Value |
|---|---|
| Optimizer for W,G,V | SGD |
| Initial learning rate for W,G,V | 0.1 |
| Learning rate scheduler for W,G,V | Cosine decay |
| Minimum learning rate for W,G,V | 0.0 |
| Momentum for W,G,V | 0.9 |
| Weight decay for W,G,V | 0.0003 |
| Optimizer for A,B | Adam |
| Learning rate for A | 0.0006 |
| Learning rate for B | 0.0003 |
| Weight decay for A,B | 0.001 |
| Initial channels for W | 16 |
| Layers for W | 8 |
| Gradient Clip for W | 5 |
| Batch size | 256 |
| Epochs | 50 |
| $\lambda$ | 0.1 |

Table 31: Hyperparameter settings in Ours-PCDARTS on ImageNet during architecture search

| Name | Value |
|---|---|
| Optimizer for W,G,V | SGD |
| Initial learning rate for W,G,V | 0.5 |
| Learning rate scheduler for W,G,V | Cosine decay |
| Minimum learning rate for W,G,V | 0.0 |
| Momentum for W,G,V | 0.9 |
| Weight decay for W,G,V | 0.0003 |
| Optimizer for A,B | Adam |
| Learning rate for A,B | 0.006 |
| Weight decay for A,B | 0.001 |
| Initial channels for W | 16 |
| Layers for W | 8 |
| Gradient Clip for W,G,V | 5 |
| Batch size | 768 |
| Epochs | 50 |
| $\lambda$ | 1 |

Table 32: Hyperparameter settings in Ours-PRDARTS on CIFAR-10/100 during architecture search

| Name | Value |
|---|---|
| Optimizer for W,G,V | SGD |
| Initial learning rate for W,G,V | 0.025 |
| Learning rate scheduler for W,G,V | Cosine decay |
| Minimum learning rate for W,G,V | 0.001 |
| Momentum for W,G,V | 0.9 |
| Weight decay for W,G,V | 0.0003 |
| Optimizer for A,B | Adam |
| Learning rate for A,B | 0.0003 |
| Weight decay for A,B | 0.001 |
| Initial channels for W | 16 |
| Layers for W | 8 |
| Gradient Clip for W | 5 |
| Batch size | 64 |
| Epochs | 50 |
| $\lambda$ | 1 |

Table 33: Hyperparameter settings in Ours-DARTS, Ours-DARTS$^{+}$ and Ours-DARTS$^{-}$ on CIFAR-10/100 during architecture evaluation

| Name | Value |
|---|---|
| Optimizer | SGD |
| Initial learning rate | 0.025 |
| Learning rate scheduler | Cosine decay |
| Momentum | 0.9 |
| Weight decay | 0.0003 |
| Initial channels | 36 |
| Layers | 20 |
| Auxiliary weight | 0.4 |
| Cutout length | 16 |
| Drop path prob | 0.2 |
| Gradient Clip | 5 |
| Batch size | 96 |
| Epochs | 600 |

Table 34: Hyperparameter settings in Ours-PDARTS and Ours-PCDARTS on CIFAR-10/100 during architecture evaluation

| Name | Value |
|---|---|
| Optimizer | SGD |
| Initial learning rate | 0.025 |
| Learning rate scheduler | Cosine decay |
| Momentum | 0.9 |
| Weight decay | 0.0003 |
| Initial channels | 36 |
| Layers | 20 |
| Auxiliary weight | 0.4 |
| Cutout length | 16 |
| Drop path prob | 0.2 |
| Gradient Clip | 5 |
| Batch size | 128 |
| Epochs | 600 |

Table 35: Hyperparameter settings in Ours-PRDARTS on CIFAR-10/100 during architecture evaluation

| Name | Value |
|---|---|
| Optimizer | SGD |
| Initial learning rate | 0.025 |
| Learning rate scheduler | Cosine decay |
| Momentum | 0.9 |
| Weight decay | 0.0003 |
| Initial channels | 36 |
| Layers | 20 |
| Auxiliary weight | 0.4 |
| Cutout length | 16 |
| Drop path prob | 0.2 |
| Gradient Clip | 5 |
| Batch size | 96 |
| Epochs | 600 |

Table 36: Hyperparameter settings on ImageNet during architecture evaluation

| Name | Value |
|---|---|
| Optimizer | SGD |
| Initial learning rate | 0.5 |
| Learning rate scheduler | Cosine decay |
| Momentum | 0.9 |
| Weight decay | 0.00003 |
| Initial channels | 48 |
| Layers | 14 |
| Auxiliary weight | 0.4 |
| Label smooth | 0.1 |
| Drop path prob | 0.0 |
| Gradient Clip | 5 |
| Batch size | 1024 |
| Epochs | 250 |

## F  Visualization of searched architectures

We visualize the architectures searched by our methods in Figure 11 to Figure 19.

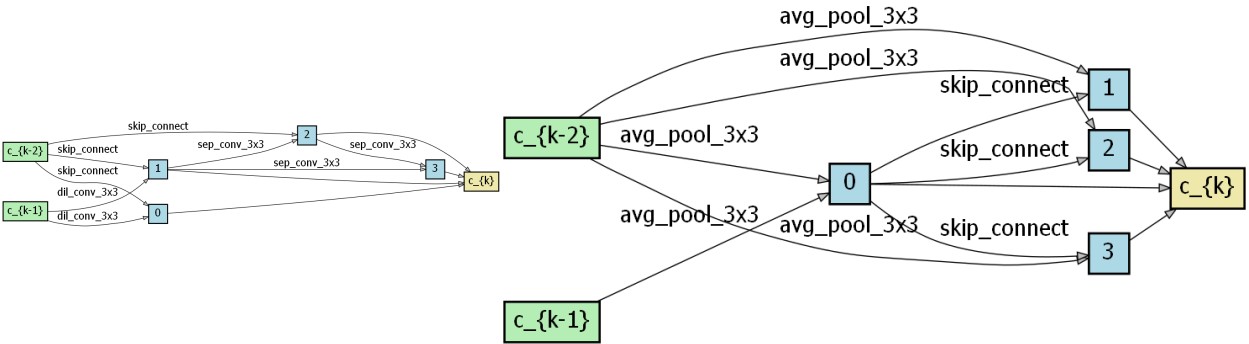

Figure 11: Architectures searched by Ours-DARTS2nd on CIFAR-10.

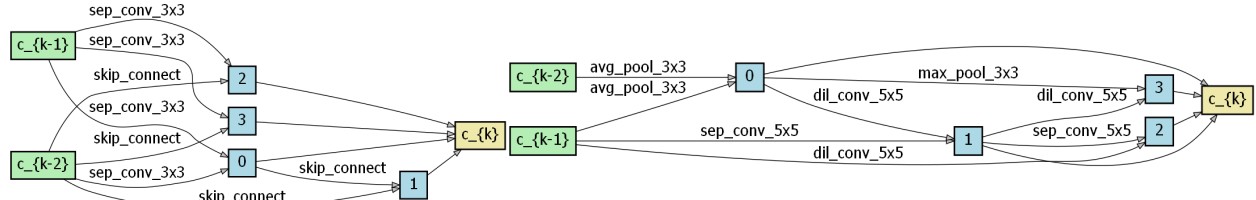

Figure 12: Architectures searched by OUrs-DARTS2nd on CIFAR-100.

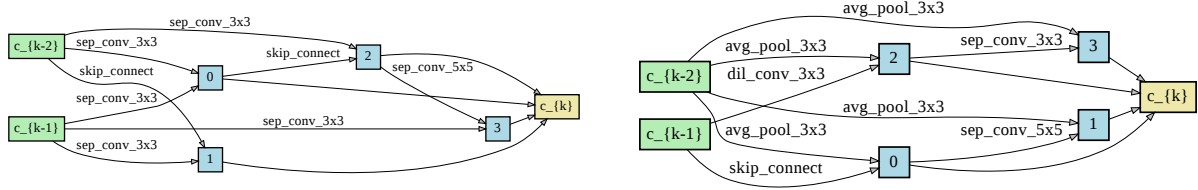

Figure 13: Architectures searched by Ours-PDARTS on CIFAR-10.

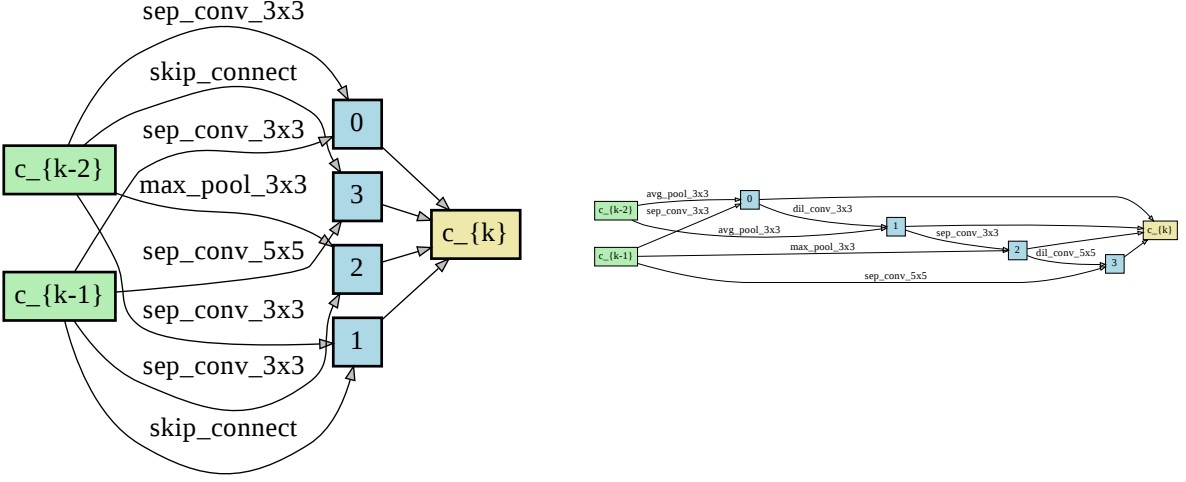

Figure 14: Architectures searched by Ours-PDARTS on CIFAR-100.

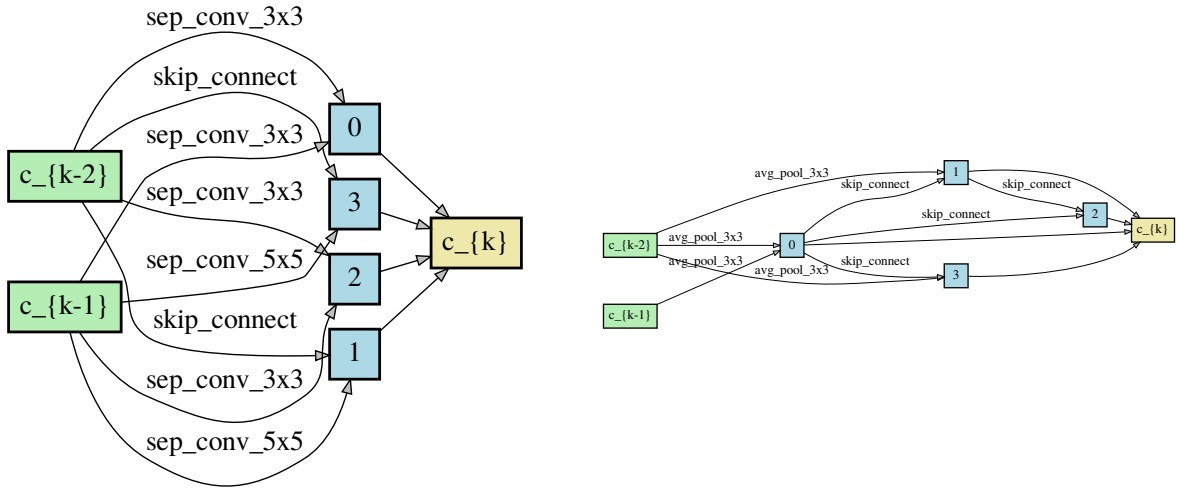

Figure 15: Architectures searched by Ours-PCDARTS on CIFAR-10.

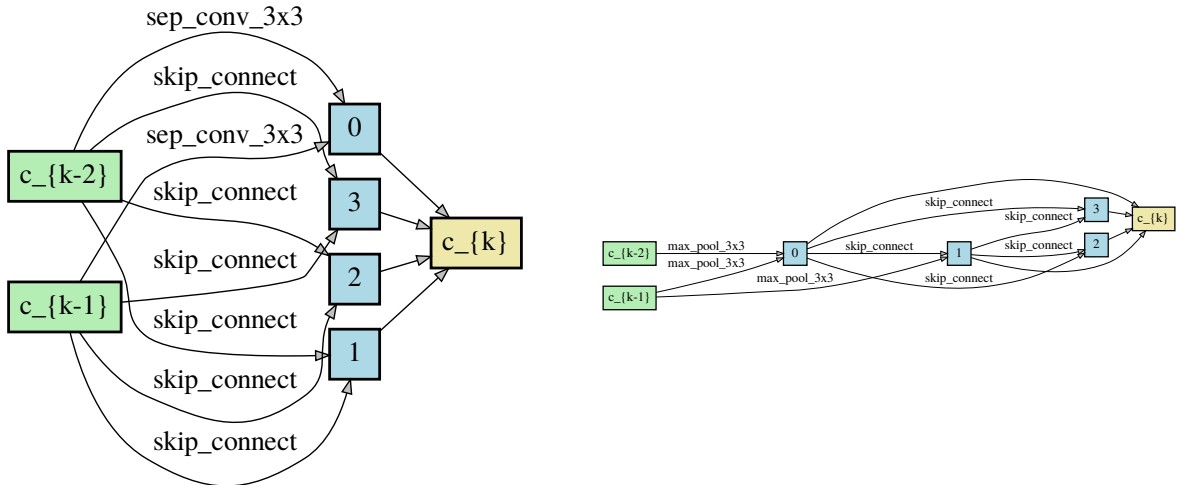

Figure 16: Architectures searched by Ours-PCDARTS on CIFAR-100.

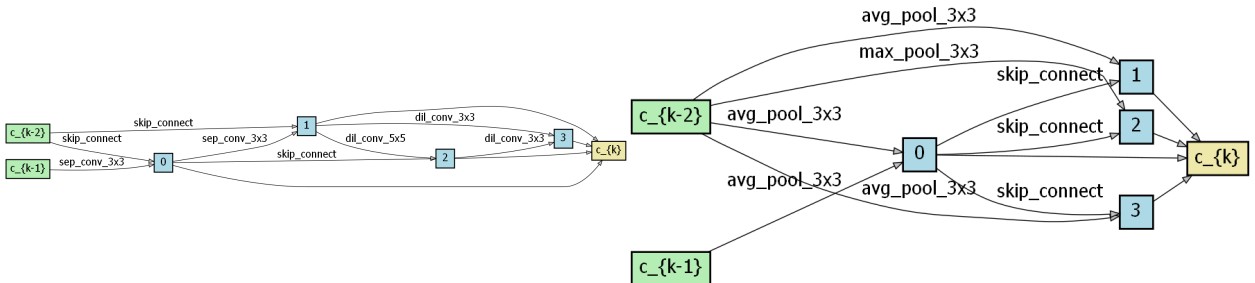

Figure 17: Architectures searched by Ours-PRDARTS on CIFAR-10.

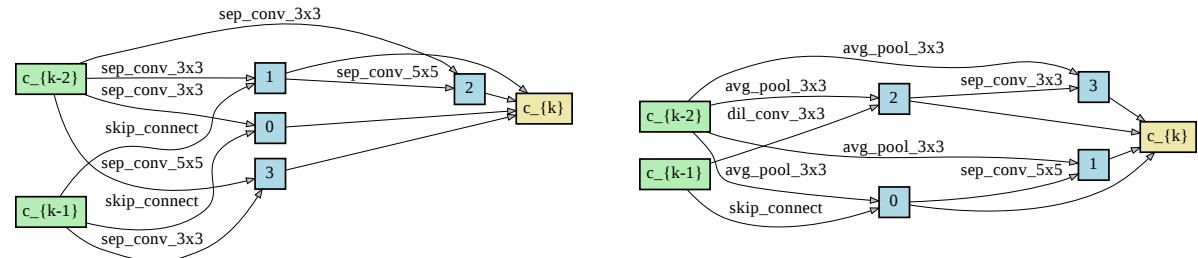

Figure 18: Architectures searched by Ours-PRDARTS on CIFAR-100.

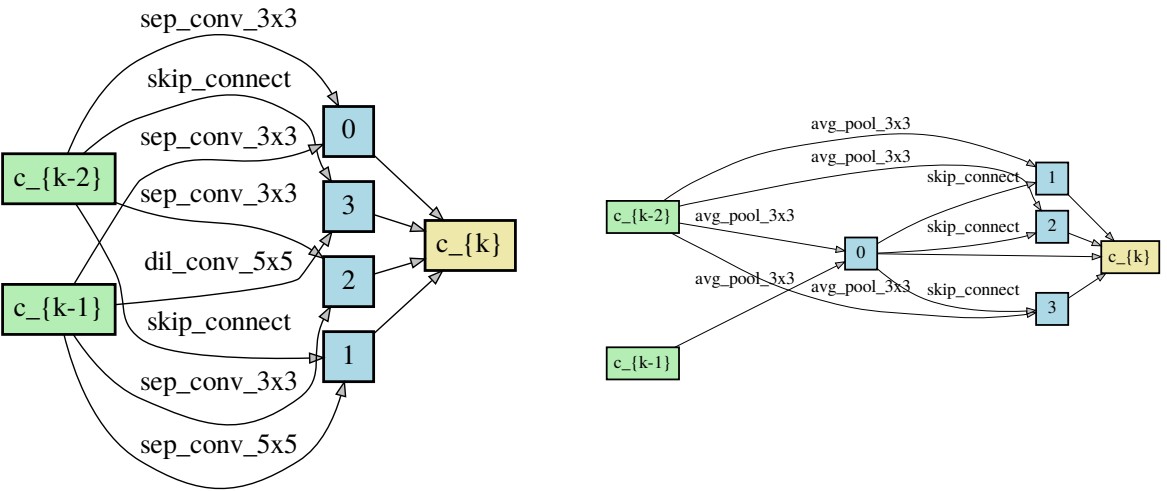

Figure 19: Architectures searched by Ours-PCDARTS on ImageNet (directly searched).

# G  Additional related works

**Adversarial learning.**  Adversarial learning (Goodfellow et al., 2014a) has been widely applied to 1) data generation (Goodfellow et al., 2014a; Yu et al., 2017) where a discriminator tries to distinguish between generated images and real images and a generator is trained to generate realistic data by making such a discrimination difficult to achieve; 2) domain adaptation (Ganin & Lempitsky, 2015) where a discriminator tries to differentiate between source images and target images while the feature learner learns representations which make such a discrimination unachievable; 3) adversarial attack and defence (Goodfellow et al., 2014b) where an attacker adds small perturbations to the input data to alter the prediction outcome and the defender trains the model in a way that the prediction outcome remains the same given perturbed inputs.

**Bi-level optimization (BLO).**  BLO (Dempe, 2002) has been broadly applied for hyperparameter tuning (Feurer et al., 2015), neural architecture search (Liu et al., 2019), meta learning (Finn et al., 2017), data reweighting (Shu et al., 2019; Ren et al., 2020; Wang et al., 2020), learning rate adjustment (Baydin et al., 2017), label denoising (Zheng et al., 2019), data generation (Such et al., 2019). In these methods, meta parameters (e.g, hyperparameters, neural architectures, data weights, etc.) are optimized by minimizing validation losses in an upper-level optimization problem and model weights are learned by minimizing training losses in a lower-level optimization problem. Pessimistic bi-level optimization (Dempe, 2002; Dempe et al., 2014; Liu et al., 2021) solves a min-max problem where maximization is conducted on lower-level optimization variables and minimization is conducted on upper-level optimization variables. Our work is different from pessimistic bi-level optimization. In our work, lower-level optimization variables (i.e., model weights) are minimized instead of being maximized; for upper-level optimization variables (i.e., architectures, hyperparameters), some of them are maximized while others are minimized.

