# OpenReview forum: "Improving Neural Architecture Search by Minimizing Worst-Case Validation Loss"
_TMLR — Rejected by TMLR_

### Review · Reviewer_2gKj · 2024-06-30

**Summary Of Contributions:**

Summary

1. The paper proposes a new method to evaluate and improve the worst-case performance of a neural architecture, which is important in practical applications where worst-case performance can have significant consequences (e.g., in a treatment recommendation system).

2. The method uses multi-level optimization, which consists of four levels of nested optimization problems to train the model.

3. The framework has the potential to be used in various applications where worst-case performance is critical.

**Audience:**

Yes

**Broader Impact Concerns:**

While the approach optimizes for the worst case scenario, the resulted model may not consider the overall demographics of the data which leads to skewed bias towards certain distribution. This may potentially lead to ethical issues in human society.

**Claims And Evidence:**

No

**Requested Changes:**

A discussion about how the data noise influences the overrall performance of the system would be helpful to justify the robustness of the approach as minimax approach is known to be less robust in terms of optimization.

**Strengths And Weaknesses:**

Pros:

1. The concept of leveraging neural networks for data generation has promises.
2. Optimizing for the worst-case scenario ensures robustness and reliability in the system, making it more effective in a wide range of contexts.

Cons:

1. The proposed approach demonstrates a limited level of innovation.
2. The potential for noise or errors in generated data is a significant concern, compromising integrity and validity if not properly addressed.
3. State 1 and stage 2 seem not sequentially related. They can happen in parallel. Not sure if describing them as multi-stage the best way.
4. The performance guarantees provided for the generator require further validation and evidence to instill confidence in its ability to consistently produce high-quality data.

---

### Review · Reviewer_q8cJ · 2024-07-08

**Summary Of Contributions:**

The submission presents a novel multi-level optimization framework aimed at improving the worst-case performance of neural architectures. Key contributions include:

Adversarial Validation Examples: The framework employs a tester model to generate adversarial validation examples, which are used to optimize the learner model's architecture by minimizing the loss on these challenging examples.

Robustness and Efficiency: Extensive experiments demonstrate the method's effectiveness in improving robustness against adversarial attacks and reducing search and memory costs compared to baseline methods.

Comprehensive Evaluation: The framework's performance is validated on various datasets (CIFAR-10, CIFAR-100, ImageNet) and under different adversarial attacks, showing significant improvements in both robustness and accuracy.

**Audience:**

Yes

**Claims And Evidence:**

Yes

**Requested Changes:**

Detailed Applicability: Provide a more detailed discussion on the limitations regarding non-differentiable NAS methods and potential ways to mitigate these issues. This change is critical for broader applicability.

Simplify Implementation: Include more detailed guidelines or tools to simplify the implementation of the multi-level optimization process. This would strengthen the work by making it more accessible to practitioners.

Additional Ablation Studies: Conduct and present more ablation studies to isolate the effects of individual components of the framework. This would enhance the understanding of the method's contributions and is important for securing acceptance.

**Strengths And Weaknesses:**

Strengths:

Innovative Framework: The use of a multi-level optimization approach to enhance worst-case performance is novel and effective.

Extensive Experiments: The submission includes comprehensive experiments on multiple datasets, demonstrating the robustness and efficiency of the proposed method.

Clear Impact: The framework shows clear improvements in adversarial robustness and generalization across datasets.

Weaknesses:

Applicability: The method may be challenging to apply to non-differentiable NAS methods, such as those based on reinforcement learning or evolutionary algorithms.

Complexity: The multi-level optimization process and the need for hyperparameter tuning may increase the complexity of implementation and use.

---

### Review · Reviewer_cAZm · 2024-08-12

**Summary Of Contributions:**

This paper proposes to tackle Neural architecture search (NAS) from a perspective of worst-case validation loss. Specifically, it presents a framework that incorporates a learner model and a tester model for evaluating the worst-case performance of machine learning models. The tester model utilizes a deep generative model to create generative adversarial validation examples that challenge the learner's performance. Meanwhile, the learner retrains to improve its worst-case performance based on the generated validation examples. The framework involves multi-level optimization stages. The experiments show the effectiveness of the proposed framework on different benchmarks.

**Audience:**

Yes

**Broader Impact Concerns:**

None.

**Claims And Evidence:**

Yes

**Requested Changes:**

1. Please provide more insights of this worst-case performance design.
2. Please discuss the connection between Eq.4 and adversarial training. And clarify the reason why Eq. 4 can improve the model performance on clean data.
3. Please clarify the computational costs.
4. Please include more comparisons with recent NAS algorithms.

**Strengths And Weaknesses:**

Strengths:
1. The paper is well-organized and easy to follow.
2. The generation of adversarial validation examples to evaluate worst-case performance for a better searching in NAS seems interesting.
3. The authors conduct extensive experiments with different NAS baselines on different datasets.

Weaknesses:
1. Although the adversarial validation examples seem interesting, it is difficult to see the reason why the worst-case performance of models should be considered as an important component during searching in NAS. More insights of this design and some empirical evidence are recommended.
2. According to Eq. 4, the generated validation examples can be treated as some special adversarial examples without norm constraint. However, it is well-known that the adversarial training could harm the model performance on clean data. Why the min-max optimization in Eq. 4 helps the performance of model on clean data remains unclear.
3. The framework requires multi-level optimization stages, which should require more computational costs. However, according to Table 5, the proposed framework achieves the same searching cost as the baseline, which needs clarification.
4. The compared algorithms are before 2020. The comparison should include some recent state-of-the-art algorithms, such as [a,b,c,d,e]

[a]. How Powerful are Performance Predictors in Neural Architecture Search? NeurIPS 2021.

[b]. MIGO-NAS: Towards Fast and Generalizable Neural Architecture Search. TPAMI 2021.

[c]. Neural architecture search via proxy validation. TPAMI 2022.

[d]. Renas: Relativistic evaluation of neural architecture search. CVPR 2021.

[e]. ZARTS: On Zero-order Optimization for Neural Architecture Search. NeurIPS 2022.

---

### Review · Reviewer_fkRv · 2024-08-19

**Summary Of Contributions:**

This paper proposes a neural architecture search (NAS) method that generates adversarial validation examples for improving the worst-case validation performance. The proposed method leverages a deep generative model to generate adversarial examples and is formulated as a multi-level optimization problem. The effectiveness of the proposed method is validated on several image classification datasets, including CIFAR-10, 100, and ImageNet.

**Audience:**

Yes

**Broader Impact Concerns:**

I do not have an additional concern about the ethical implications of this paper.

**Claims And Evidence:**

Yes

**Requested Changes:**

1. Several searched architectures illustrated in Appendix F contain many skip connections and pooling operations, such as in Figure 16. As these operations have no learnable parameters, the reviewer cannot understand why these architectures perform well.
1. The justification and contribution of the architecture search for improving worst-case performance seems not to be well explained. The reviewer feels that the weight parameters can be trained to improve the worst-case performance by using the proposed loss functions.
1. The authors mention, "Our framework can be used together with any differentiable search method." In the experiments, the proposed method is combined with several continuous relaxation-based architecture search methods. Is it possible to combine the proposed method with stochastic relaxation-based architecture search methods?
1. In Section 4.2, the authors mention, "the formulation in Eq. (5) is applied to DARTS and the formulation in Eq. (6) is applied to P-DARTS, PC-DARTS, and PR-DARTS." Could you provide the reason for this choice?
1. In the proposed method, three architectures exist. It might be a possible option to list or illustrate the role of these architectures and notation in the main text for readability.
1. The proposed method is only applied to image classification tasks. It might be better to explain the generality of the proposed method. Is it possible to use it for other tasks and modalities?

**Minor comments**
1. In the sentence "In this table, we also..." in the second paragraph on page 7, which table does the author indicate?
1. In Table 4, the reviewer cannot identify the correspondence between symbols like S1 and S2 and specific search spaces.

**Strengths And Weaknesses:**

[Strengths]
- Considering the worst-case performance of neural architectures in the NAS scenario is novel.
- The experimental comparison and evaluation, including ablation study and hyperparameter sensitivity analysis, evidence the effectiveness of the proposed method. The proposed method can improve not only the worst-case performance but also the average performance.
- The detailed experimental settings and additional experimental results are provided in the appendix.

[Weaknesses]
- The proposed algorithm is somewhat complicated and has many components, which seem to include a relatively large number of hyperparameters.
- The key components of the proposed method are somewhat confusing, meaning that it would be better to clarify the main technical contribution of the paper.
- The searched architectures seem to be strange (please see the first question of Requested Changes in detail).

---

### Decision · Action_Editor_G3NJ · 2024-10-02

**Recommendation:** Reject

**Comment:**

The reviewers appreciated the main contributions but were concerned about the complexity of the multi-level approach and some design choices. Unfortunately, the authors did not provide any response to any of the reviewers' questions, resulting in a universal opinion that the manuscript is not ready for publication at TMLR. We encourage the authors to read the reviewers' comments and address them in the next version.

**Audience:**

Yes, neural architectural search is important and most reviewers found the multi-level approach interesting.

**Claims And Evidence:**

The authors conducted extensive experiments to validate their claims. However, the reviewers were concerned about the complexity and some design choices, which are not very well-explained or supported yet.

**Resubmission Of Major Revision:**

The authors may consider submitting a major revision at a later time.